# σ[28]-dependent small RNA regulation of flagella biosynthesis

**Sahar Melamed[1,2]\*, Aixia Zhang[1], Michal Jarnik[1], Joshua Mills[1†], Aviezer Silverman[2], Hongen Zhang[3], Gisela Storz[1]\***

[1]Division of Molecular and Cellular Biology, Eunice Kennedy Shriver National Institute of Child Health and Human Development, Bethesda, United States; [2]Department of Microbiology and Molecular Genetics, Institute for Medical Research Israel-Canada, Faculty of Medicine, The Hebrew University of Jerusalem, Jerusalem, Israel; [3]Bioinformatics and Scientific Computing Core, Eunice Kennedy Shriver National Institute of Child Health and Human Development, Bethesda, United States

**\*For correspondence:**
sahar.melamed@mail.huji.ac.il
(SM);
storzg@mail.nih.gov (GS)

**Present address:** †Max Planck Institute for Marine Microbiology, Bremen, Germany

**Competing interest:** The authors declare that no competing interests exist.

**Abstract** Flagella are important for bacterial motility as well as for pathogenesis. Synthesis of these structures is energy intensive and, while extensive transcriptional regulation has been described, little is known about the posttranscriptional regulation. Small RNAs (sRNAs) are widespread posttranscriptional regulators, most base pairing with mRNAs to affect their stability and/or translation. Here, we describe four UTR-derived sRNAs (UhpU, MotR, FliX and FlgO) whose expression is controlled by the flagella sigma factor σ[28] (*fliA*) in *Escherichia coli*. Interestingly, the four sRNAs have varied effects on flagellin protein levels, flagella number and cell motility. UhpU, corresponding to the 3′ UTR of a metabolic gene, likely has hundreds of targets including a transcriptional regulator at the top flagella regulatory cascade connecting metabolism and flagella synthesis. Unlike most sRNAs, MotR and FliX base pair within the coding sequences of target mRNAs and act on ribosomal protein mRNAs connecting ribosome production and flagella synthesis. The study shows how sRNA-mediated regulation can overlay a complex network enabling nuanced control of flagella synthesis.

## eLife assessment

This article provides **important** findings on how bacteria use small RNAs to regulate flagellar expression with implications for multiple fields. The data supporting the conclusions are **convincing** with a large amount of data that include results from phenotypic analyses, genomics approaches as well as in-vitro and in-vivo target identification and validation methods. This study on the varied effects of three sRNAs (UhpU, FliX and MotR) is of broad interest to RNA biochemists and microbiologists.

## Introduction

Most bacteria are motile and can swim through liquid and semiliquid environments in large part driven by the flagellum. The highly complex bacterial flagellum consists of three major domains: an ion-driven motor, which can provide torque in either direction; a universal joint called the hook-basal body, which transmits motor torque; and a 20-nm-thick hollow filament tube composed of the flagellin subunit, which acts as a propeller (reviewed in *Altegoer and Bange, 2015*; *Nakamura and Minamino, 2019*). The complete flagellum is comprised of many proteins, and the flagellar regulon encompasses more than 50 genes. Flagella are costly for the cell to synthesize, requiring up to ~2% of the cell's biosynthetic energy expenditure and extensive use of ribosomes (reviewed in *Soutourina and Bertin, 2003*; *Guttenplan and Kearns, 2013*).

To ensure that flagellar components are made in the order in which they are needed, transcription of the genes in the regulon is activated in a sequential manner in *Escherichia coli* (*Kalir et al., 2001*) and *Salmonella enterica* (reviewed in *Chevance and Hughes, 2008*). The genes can be divided into three groups based on their time of activation: early genes, middle genes, and late genes (*Figure 1A*). The FlhDC transcription regulators, encoded by the two early genes, activate the transcription of the middle genes (Class 2), which are required for the hook-basal body. FlhDC also activates transcription of *fliA*, encoding sigma factor $\sigma^{28}$ (*Fitzgerald et al., 2014*). $\sigma^{28}$ in turn activates transcription of the late genes responsible for completing the flagellum and the chemotaxis system (Class 3). $\sigma^{28}$ additionally increases expression of several of the middle genes (Class 2/3) (*Fitzgerald et al., 2014*). $\sigma^{28}$ activity itself is negatively regulated by the anti-sigma factor, FlgM, which is transported out of the cell, freeing $\sigma^{28}$, when the hook-basal body complex is complete (reviewed in *Smith and Hoover, 2009*; *Osterman et al., 2015*). Given the numerous components required at different times and in different stoichiometries during flagellum assembly, various factors can be rate limiting under specific conditions (reviewed in *Chevance and Hughes, 2008*). The dependence of flagella synthesis on FlhDC and $\sigma^{28}$ generates a coherent feed-forward loop. In this loop, the first regulator (FlhDC) activates the second regulator ($\sigma^{28}$), and they both additively activate their target genes. This results in prolonged flagellar expression, protecting the flagella synthesis from a transient loss of input signal (*Kalir et al., 2005*).

Given flagella are so costly to produce, synthesis is tightly regulated such that flagellar components are only made when motility is beneficial. Thus, flagellar synthesis is strongly impacted by environmental signals. For instance, flagellar gene expression is decreased in the presence of D-glucose, in high temperatures, high salt, and extreme pH, as well as the presence of DNA gyrase inhibitors (*Shi et al., 1993*; *Adler and Templeton, 1967*). The flagellar genes are activated under oxygen-limited conditions (*Landini and Zehnder, 2002*) and at various stages of infection (reviewed in *Erhardt, 2016*). Consequently, transcription of many genes in the flagellar regulon is regulated in response to a range of environmental signals. For example, the transcription of *flhDC* is controlled by at least 13 transcription factors, each of them active under different conditions (reviewed in *Prüß, 2017*).

While the activation of flagella synthesis has been examined in some detail, there has been less investigation into the termination of synthesis, which we presume is equally important for the conservation of resources. Additionally, while transcriptional regulation of flagella genes has been studied for many years, the post-transcriptional control of the regulon has only received limited attention. Small RNAs (sRNAs) that can originate from many different genetic loci (reviewed in *Adams and Storz, 2020*) are key post-transcriptional regulators in bacteria. They usually regulate their targets in trans via limited base-pairing, affecting translation and/or mRNA stability (reviewed in *Hör et al., 2020*; *Papenfort and Melamed, 2023*). Many characterized sRNAs are stabilized and their base pairing with targets increased by RNA chaperones, of which the hexameric, ring-shaped Hfq protein has been studied most extensively (reviewed in *Updegrove et al., 2016*; *Holmqvist and Vogel, 2018*). The only post-transcriptional control by base pairing sRNAs described for the *E. coli* flagellar regulon thus far is negative regulation of *flhDC* by ArcZ, OmrA, OmrB, OxyS (*De Lay and Gottesman, 2012*), and AsflhD (encoded antisense to *flhD*)(*Lejars et al., 2022*), positive regulation of the same mRNA by McaS (*Thomason et al., 2012*), and negative regulation of *flgM* by OmrA and OmrB (*Romilly et al., 2020*). These sRNAs and a few other sRNAs also were shown to affect motility and biofilm formation (*Bak et al., 2015*).

In this study, we characterized four $\sigma^{28}$-dependent sRNAs, which were detected with their targets on Hfq through RIL-seq methodology that captures the sRNA-target interactome (*Melamed et al., 2016*; *Melamed et al., 2020* and reviewed in *Silverman and Melamed, 2023*). These sRNAs originate from the untranslated regions (UTRs) of mRNAs, three of which belong to the flagellar regulon. We identified a wide range of targets for the sRNAs, including genes related to flagella and ribosome synthesis and observed that the sRNAs act on some of these targets by unique modes of action. We also found that three of these sRNAs regulate flagella number and bacterial motility, possibly imposing temporal control on flagella synthesis and integrating metabolic signals into this complex regulatory network.

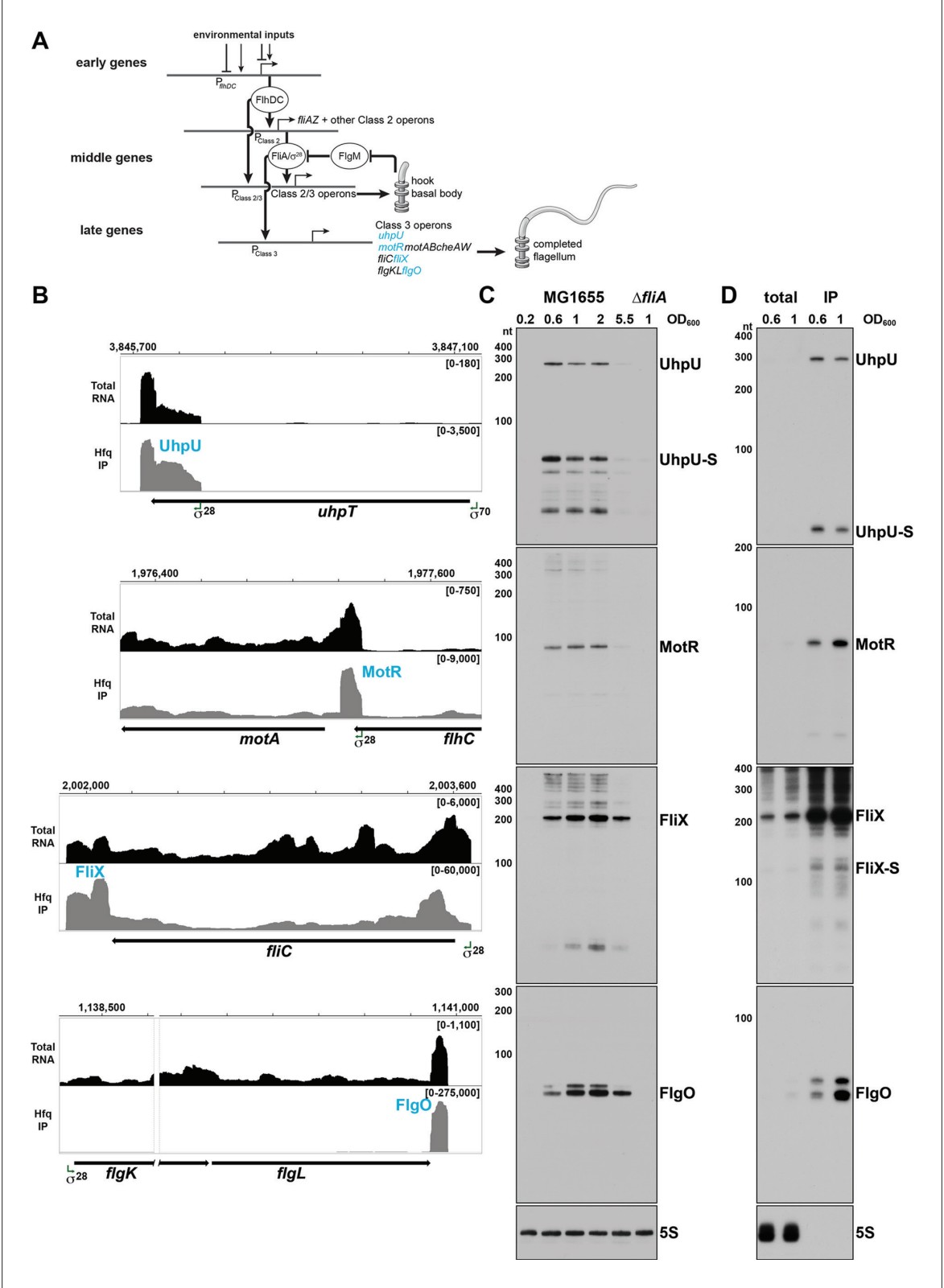

**Figure 1.** σ28-Dependent sRNAs are primarily expressed in log phase. (**A**) Overview of the flagellar regulon. The early genes initiate the transcription of the middle genes, including *fliA* which encodes σ28. In turn, σ28 initiates the transcription of the late genes and enhances the transcription of some of the middle genes. For the middle and late genes, only selected operons are shown. The sRNAs analyzed in this study are colored in blue. This model was inspired by ***Kalir et al., 2005***. (**B**) Browser images showing levels of UhpU, MotR, FliX, and FlgO sRNAs in total RNA (black) and Hfq

*Figure 1 continued on next page*

*Figure 1 continued*

co-immunoprecipitation (gray) libraries. Normalized read count ranges are shown in the upper right of each frame. Data analyzed is from (RIL-seq experiment 1, *Melamed et al., 2020*). (C) Northern blot analysis of total RNA from WT (GSO983) or Δ*fliA* (GSO1068) cells grown to the indicated time points. A full-length transcript (~260 nt) and several processed transcripts, of which one is predominant (UhpU-S,~60 nt), are detected for UhpU, one prominent band (~95 nt) is detected for MotR, one prominent band (~200 nt) is detected for FliX, and two bands close in size (~75 nt) are detected for FlgO. (D) Northern blot analysis of WT (GSO983) cells grown to OD$_{600}$ ~0.6 and~1.0. RNA was extracted from total lysates as well as samples from co-immunoprecipitation with Hfq, separated on an acrylamide gel, transferred to a membrane, and probed for σ$^{28}$-dependent sRNAs. A~100 nt FliX band (FliX-S) was revealed immunoprecipitating with Hfq. In (C) and (D), RNAs were probed sequentially on the same membrane, and the 5S RNA served as a loading control.

The online version of this article includes the following figure supplement(s) for figure 1:

**Figure supplement 1.** Sequences and predicted structures of UhpU, MotR, FliX, and FlgO sRNAs and effect of carbon source on sRNA levels.

**Figure supplement 2.** UhpU, MotR, FliX and FlgO levels across growth.

## Results

### σ$^{28}$-dependent sRNAs are expressed sequentially in log phase cells

Analysis of several different RNA-seq data sets suggested the expression of four σ$^{28}$-dependent sRNAs in *E. coli*. σ$^{28}$-dependent expression of the sRNAs was detected using ChIP-seq and RNA-seq in a comprehensive analysis of the σ$^{28}$ regulon (*Fitzgerald et al., 2014*), while the position and nature of the 5′ ends were revealed by a 5′ end mapping study (*Thomason et al., 2015*). Regulatory roles were indicated by binding to other RNAs in RIL-seq data (*Melamed et al., 2016*; *Melamed et al., 2020*; *Bar et al., 2021*). The four sRNAs originate from the UTRs of protein coding genes (*Figure 1B* and *Figure 1—figure supplement 1A*). UhpU corresponds to the 3′ UTR of *uhpT*, which encodes a hexose phosphate transporter (*Marger and Saier, 1993*). UhpU is transcribed from its own promoter inside the coding sequence (CDS) of *uhpT* (*Thomason et al., 2015*). The other three σ$^{28}$-dependent sRNAs correspond to the UTRs of the late genes in the flagellar regulon. MotR originates from the 5′ UTR of *motA*, which encodes part of the flagellar motor complex. Based on previous transcription start site analysis, the promoter for *motR* is within the *flhC* CDS and is also the promoter of the downstream *motAB-cheAW* operon (*Thomason et al., 2015*; *Fitzgerald et al., 2014*). FliX originates from the 3′ UTR of *fliC,* which encodes flagellin, the core component of the flagellar filament (reviewed in *Thomson et al., 2018*). FlgO originates from the 3′ UTR of *flgL*, a gene that encodes a junction protein shown to connect the flagella to the hook in *S. enterica* (*Ikeda et al., 1987*). The observation that FliX and FlgO levels decline substantially in RNA-seq libraries treated with 5′ phosphate-dependent exonuclease to deplete processed RNAs (*Thomason et al., 2015*), indicates that both of these sRNAs are processed from their parental mRNAs.

Northern blot analysis confirmed σ$^{28}$-dependent synthesis of these sRNAs since expression was significantly decreased in a mutant lacking σ$^{28}$ (Δ*fliA*) (*Figure 1C*). Given that most σ$^{28}$-dependent mRNAs encode flagella components, the regulation suggests the sRNAs impact flagella synthesis. The northern analysis also showed that the levels of the four σ$^{28}$-dependent sRNAs are highest in the transition from mid-exponential to stationary phase growth, though there are some differences with UhpU and MotR peaking before FliX and FlgO (*Figure 1C* and *Figure 1—figure supplement 2*). Since flagellar components are expressed at precise times, the difference in the UhpU and MotR peak times compared to the FliX and FlgO peak times hints at different roles for each of these sRNAs. For UhpU, two predominant bands were observed, a long transcript and a shorter transcript processed from UhpU (denoted UhpU-S), which corresponds to the higher peak in the sequencing data (*Figure 1B*). One prominent band was detected for MotR and for FliX, while a doublet was observed for FlgO. Additional higher bands detected by the MotR probe could be explained by RNA polymerase read-through of the MotR terminator into the downstream *motAB-cheAW* operon, while the additional bands seen for FliX could be explained by alternative processing of the *fliC* mRNA.

We also examined the levels of the four sRNAs in minimal media (M63) supplemented with different carbon sources (*Figure 1—figure supplement 1B*). Generally, the sRNAs levels in minimal medium are comparable to or slightly higher to the levels in rich media (LB) except in medium with glucose-6-phosphate (G6P), where the levels of UhpU-S are significantly elevated while the levels of full-length UhpU transcript and the other σ$^{28}$-dependent sRNAs are decreased. These observations suggest an alternative means for UhpU-S generation from the *uhpT* mRNA known to be induced by G6P (*Postma*

*et al., 2001*). We also observe more FliX products, particularly for cells grown in minimal medium with ribose or galactose.

The predicted structures for the four σ28-dependent sRNAs (*Figure 1—figure supplement 1C*), with strong stem-loops at the 3′ ends, are consistent with the structures of known Hfq-binding sRNAs and the association with Hfq observed in the RIL-seq data (*Melamed et al., 2016*). To confirm Hfq binding, we probed RNA that co-immunoprecipitated with Hfq (*Figure 1D*). Strong enrichment and fewer background bands were observed for all of the sRNAs; ~260 nt and ~60 nt bands for UhpU and UhpU-S, respectively, a~95 nt band for MotR, a ~200 nt band for FliX and a doublet of ~75 nt bands for FlgO. For FliX, we also detected a second ~100 nt FliX band (denoted FliX-S; *Figure 1—figure supplement 1A*) that corresponds to the 3′ peak in the sequencing data (*Figure 1B*) and includes one of the repetitive extragenic palindromic (REP) sequences downstream of *fliC*.

## σ28-dependent sRNAs impact flagella number and bacterial motility

To begin to decipher the roles of the four σ28-dependent sRNAs, we constructed plasmids for over-expression of the sRNAs (*Figure 2—figure supplement 1A*). Given that it was challenging to obtain constructs constitutively overexpressing UhpU because all clones had mutations, this sRNA could only be expressed from a plasmid when controlled by an IPTG-inducible $P_{lac}$ promoter (*Guo et al., 2014*), hinting at a critical UhpU role in *E. coli* vitality. The other sRNAs were expressed from a plasmid with the constitutive $P_{LlacO-1}$ promoter (*Urban and Vogel, 2007*). We also obtained a plasmid constitutively overexpressing MotR*, a more abundant derivative of MotR identified by chance (TGC at positions 6–8 mutated to GAG; *Figure 1—figure supplement 1A*).

We tested the effects of overexpressing the sRNAs on flagellar synthesis by determining the number of flagella by electron microscopy (EM) and on bacterial motility by assaying the spread of cells on 0.3% agar plates. The WT *E. coli* strain used throughout the paper is highly motile due to an IS1 insertion in the *crl* gene (*crl*−), thus eliminating expression of a protein that promotes σS binding to the RNA polymerase core enzyme (*Typas et al., 2007*), and resulting in higher expression of the flagellar regulatory cascade (*Pesavento et al., 2008*). However, we also assayed a less motile strain with the restored *crl*+ gene for UhpU and MotR effects on motility, given that no effects were observed with the highly motile *crl*− strain.

Intriguingly, overexpression of the individual sRNAs had different consequences. UhpU over-expression caused a slight increase in flagella number (*Figure 2A*) and a marked increase in motility (*Figure 2B*). Overexpression of MotR, particularly MotR*, led to a dramatic increase in the flagella number (*Figure 2C* and *Figure 2—figure supplement 2A*) and MotR but not MotR* had a slight effect on motility (*Figure 2D* and *Figure 2—figure supplement 2B*). It has been suggested that the run/tumble behavior of bacteria, which affect their swimming, is only weakly dependent on number of flagella (*Mears et al., 2014*), possibly explaining these somewhat contradictory effects on flagella number and motility. In contrast to UhpU and MotR, FliX overexpression led to a reduction in the number of flagella (*Figure 2E*), an effect that was even more pronounced in a strain overexpressing FliX-S (*Figure 2—figure supplement 2C*). Overexpression of FliX-S but not FliX also reduced bacterial motility (*Figure 2F* and *Figure 2—figure supplement 2D*). While FliX-S overexpression seems to lead to aflagellated bacteria, we hypothesize that the sRNA is delaying but not eliminating flagella gene expression, explaining why the bacteria are still moderately motile. Some motility phenotypes can be explained by differences in growth rate, but we do not think that this is the case for MotR and FliX as we observed only slight effects on growth upon MotR, MotR*, FliX and FliX-S overexpression (*Figure 2—figure supplement 1B*). FlgO overexpression did not result in detectable changes in our assays (*Figure 2G* and *Figure 2H*). Together, these results show that the σ28-dependent sRNAs have a range of effects on flagella number and motility, with UhpU and MotR, which are expressed first, increasing both phenotypes and FliX, which is expressed later, decreasing both. Given that MotR* and FliX-S have stronger effects for some phenotypes and provide a bigger dynamic range, these derivatives were included in subsequent assays.

## σ28-dependent sRNAs have wide range of potential targets based on RIL-seq analysis

To understand the phenotypes associated with overexpression of the σ28-dependent sRNAs, we took advantage of the sRNA-target interactome data obtained by RIL-seq (*Melamed et al., 2020*; *Melamed*

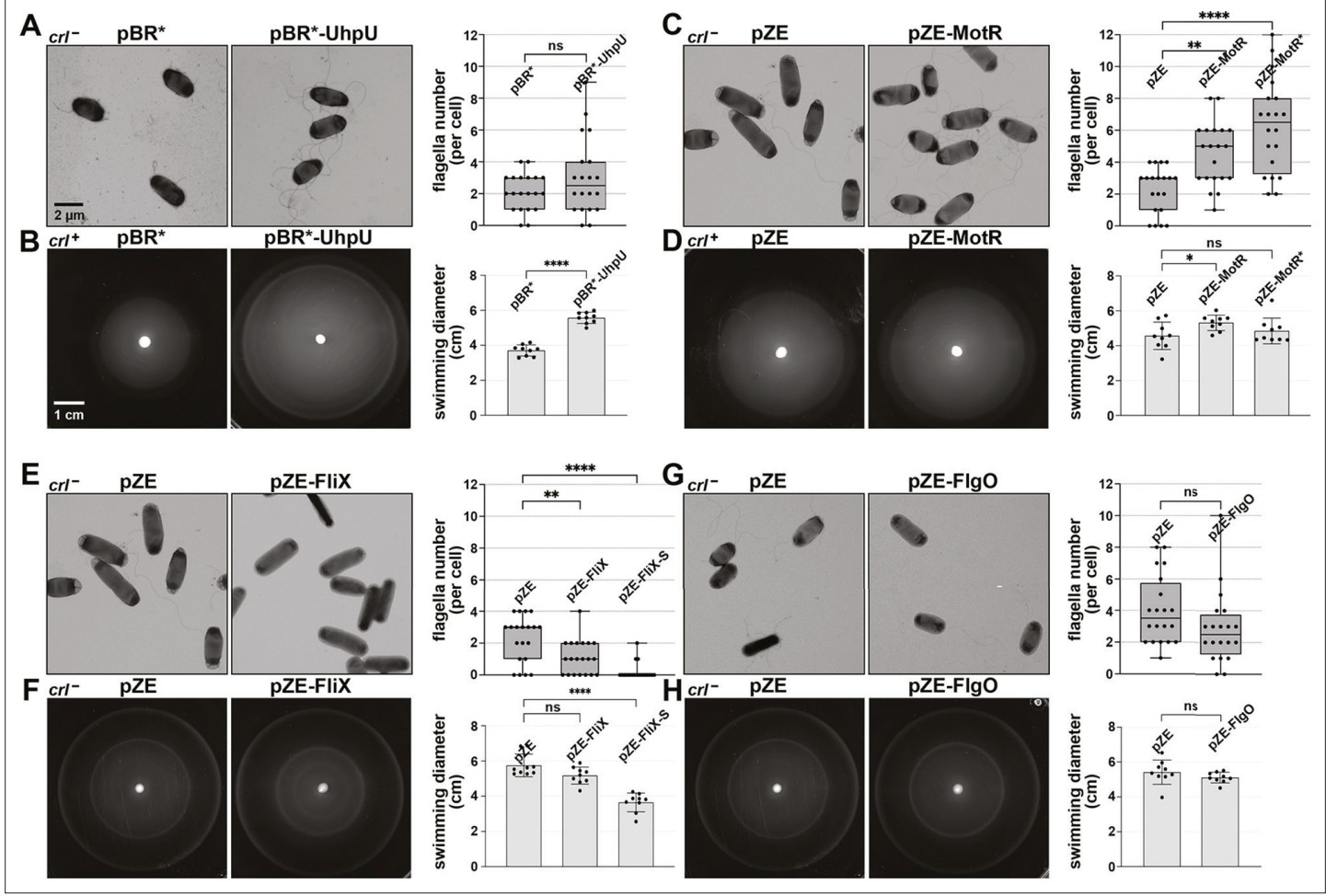

**Figure 2.** Overexpression of the σ28-dependent sRNAs leads to differences in flagella number and motility. (**A**) Moderate increase in flagella number with UhpU overexpression based on EM analysis for WT (*crl*⁻) cells carrying an empty vector or overexpressing UhpU. (**B**) Increased motility with UhpU overexpression based on motility in 0.3% agar for WT (*crl*⁺) cells carrying an empty vector or overexpressing UhpU. (**C**) Increase in flagella number with MotR overexpression based on EM analysis for WT (*crl*⁻) cells carrying an empty vector or overexpressing MotR. (**D**) Slight increase in motility with MotR overexpression based on motility in 0.3% agar for WT (*crl*⁺) cells carrying an empty vector or overexpressing MotR. (**E**) Reduction in flagella number with FliX overexpression based on EM analysis for WT (*crl*⁻) cells carrying an empty vector or overexpressing FliX. (**F**) Reduced motility with FliX overexpression based on motility in 0.3% agar for WT (*crl*⁻) cells carrying an empty vector or overexpressing FliX. (**G**) No change in flagella number with FlgO overexpression based on EM analysis for WT (*crl*⁻) cells carrying an empty vector or overexpressing FlgO. (**H**) No change in motility with FlgO overexpression based on motility in 0.3% agar for WT (*crl*⁻) cells carrying an empty vector or overexpressing FlgO. Cells in (**A**) and (**B**) were induced with 1 mM IPTG. Quantification for all the assays is shown on the right. For (**A**), (**C**), (**E**) and (**G**) quantification of the number of flagella per cell was done by counting the flagella for 20 cells (black dots), and a one-way ANOVA comparison was performed to calculate the significance of the change in flagella number (ns = not significant, **=p < 0.01, ****=p < 0.0001). Each experiment was repeated three times, and one representative experiment is shown. The bottom and top of the box are the 25th and 75th percentiles, the line inside the box is the median, the lower and the upper whiskers represent the minimum and the maximum values of the dataset, respectively. While some differences in cells size and width were observed in the EM analysis, they were not statistically significant. The experiments presented in (**C**) and (**E**) were carried out on same day, and the same pZE sample is shown. Graphs for (**B**), (**D**), (**F**) and (**H**) show the average of nine biological repeats. Error bars represent one SD, and a one-way ANOVA comparison was performed to calculate the significance of the change in motility (ns = not significant, *=p < 0.05, ****=p < 0.0001). The scales given in (**A**) and (**B**) are the same for all EM images and all motility plates, respectively.

The online version of this article includes the following figure supplement(s) for figure 2:

**Figure supplement 1.** Expression of σ28-dependent sRNAs from plasmids and the effect of MotR and FliX overexpression on growth.

**Figure supplement 2.** Effects of MotR* and FliX-S overexpression on flagella number and motility.

*et al., 2016*). We analyzed the data (*Supplementary file 1*) generated from 18 samples representing six different growth conditions, which included different stages of bacterial growth in rich medium as well as growth in minimal medium and iron-limiting conditions. We selected targets for further characterization if they were detected in the datasets for least four different conditions. The sRNAs differ significantly in their target sets (*Figure 3—figure supplement 1A*). In general, UhpU is a hub with hundreds of RIL-seq targets. Its target set comprises a wide range of genes, including multiple genes that have roles in flagella synthesis and carbon metabolism. MotR and FliX were associated with fewer targets, but intriguingly, both sets were enriched for genes encoding ribosomal proteins. We also noted that the *fliC* gene encoding flagellin was present in the target sets for UhpU, MotR, and FliX. Although FlgO is one of the most strongly enriched sRNAs upon Hfq purification (ranked fourth in *Melamed et al., 2020*), it had the smallest set of targets. Almost none of the targets were found in more than two conditions and only *gatC* was detected in four conditions, hinting FlgO might not act as a conventional Hfq-dependent base-pairing sRNA. Unlike for most characterized sRNA targets, the RIL-seq signal for the sRNA interactions with *fliC* and the ribosomal protein genes is internal to the CDSs (*Supplementary file 1* and *Figure 3—figure supplement 1B*). Before turning our attention to these unique targets, we first examined the UhpU interaction with a canonical target.

## UhpU represses expression of the LrhA transcriptional repressor of *flhDC*

We were intrigued to find that the mRNA encoding the transcription factor LrhA, which represses *flhDC* transcription, was among the top RIL-seq interactors for UhpU (*Supplementary file 1*). The signals that activate this LysR-type transcription factor (*Lehnen et al., 2002*), are not known, but the *lrhA* mRNA has an unusually long 371 nt 5′ UTR (*Figure 3A*), a feature that has been found to correlate with post-transcriptional regulation (reviewed in *Adams and Storz, 2020*). The predicted base pairing between UhpU and the *lrhA* 5′-UTR (*Figure 3B*) corresponds to the seed sequence suggested for UhpU (*Melamed et al., 2016*).

To test the effects of UhpU on this target, we fused the 5′ UTR of *lrhA*, which includes the region of the RIL-seq *lrhA*-UhpU chimeras and the predicted base-pairing region, to a *lacZ* reporter (*Mandin and Gottesman, 2009*). UhpU overexpression reduced expression of the chromosomally-encoded P_BAD-*lrhA*-*lacZ* reporter (*Figure 3C*). A single nucleotide mutation in the base pairing region of *uhpU* (*uhpU-M1*) eliminated UhpU repression of *lrhA*-*lacZ*, while a complementary mutation introduced into the chromosomal *lrhA*-*lacZ* fusion (*lrhA-M1*) restored the repression providing direct evidence for UhpU base pairing to *lrhA* leading to repression. Down-regulation of LrhA by UhpU, which is expected to lead to increased FlhDC levels, is in accord with the positive impact of UhpU on motility (*Figure 2*). To test this model, we monitored the effect of UhpU on bacterial motility in a *lrhA* deletion strain compared to a WT strain (*Figure 3D*). With UhpU overexpression, motility was increased in the WT background as expected. In contrast, while the Δ*lrhA* strain was more motile, likely due to *flhDC* de-repression, motility was unaltered by high levels of UhpU indicating that significant UhpU effects on motility are mediated by LrhA.

Interestingly, the RIL-seq data also suggested that *lrhA* directly interacts with other sRNAs such as ArcZ, RprA and McaS (*Figure 3A*). Regions of predicted base pairing overlap known seed regions for these sRNAs (*Figure 3E*). In translational reporter assays using the *lrhA*-*lacZ* fusion, both RprA and ArcZ reduced expression, while McaS, despite having the most chimeras, had no effect (*Figure 3F*). Possibly the McaS-*lrhA* interaction has other regulatory consequences such as McaS inhibition. Intriguingly, ArcZ, RprA, and LrhA form a complex regulatory network with the general stress response sigma factor σ^S encoded by *rpoS*, as previous studies showed that LrhA represses the expression of *rprA* and *rpoS* (*Peterson et al., 2006*), while ArcZ and RprA increase *rpoS* expression (reviewed in *Mika and Hengge, 2014*).

## UhpU, MotR and FliX modulate flagellin levels

The high numbers of chimeras between UhpU, MotR or FliX with the *fliC* mRNA encoding flagellin were striking, particularly between the 3′ end of *fliC* corresponding to FliX (blue) and the 5′ end of *fliC* (red) (*Figure 4A*). As mentioned above, it was also noteworthy that most of the chimeras were internal to the *fliC* CDS. When we examined the consequences of overexpressing UhpU, MotR, MotR*, FliX or FliX-S on the levels of the flagellin protein, we observed somewhat increased levels of flagellin, both

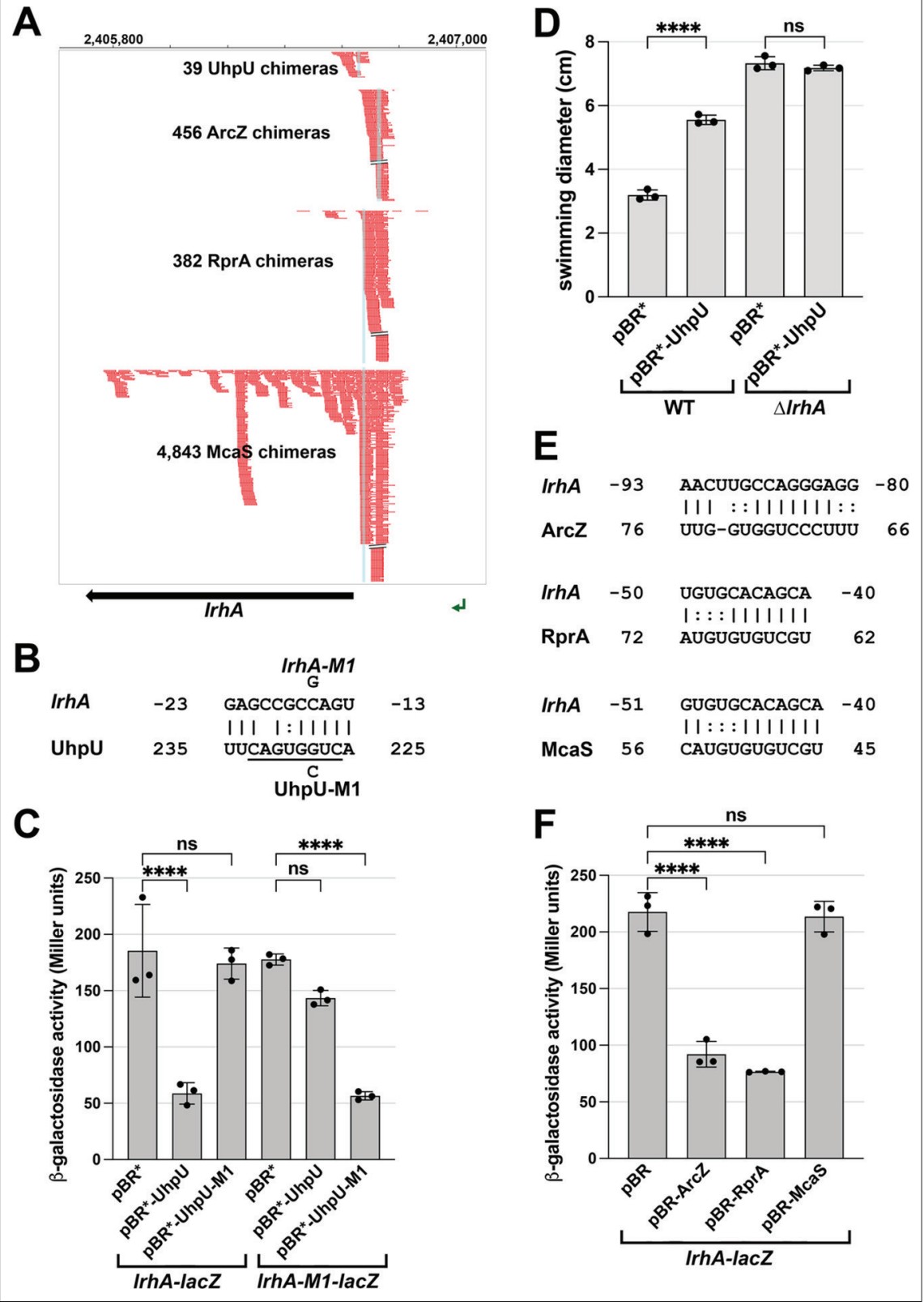

**Figure 3.** Multiple sRNAs repress LrhA synthesis. (**A**) Browser image showing chimeras (in red) for UhpU, ArcZ, RprA and McaS, at the 5′ UTR region of *lrhA*. Blue highlighting indicates position of sRNA-*lrhA* base pairing. Data analyzed is from *Melamed et al., 2020*. (**B**) Base-pairing between *lrhA* and UhpU with sequences of mutants assayed. Seed sequence predicted by *Melamed et al., 2016* is underlined. Numbering is from AUG of *lrhA* mRNA and +1 of UhpU sRNA. (**C**) UhpU represses *lrhA-lacZ* fusion based on β-galactosidase assay detecting the levels of *lrhA-lacZ* and *lrhA-M1-lacZ*

*Figure 3 continued*

translational fusions in response to UhpU and UhpU-M1 overexpression. (**D**) UhpU does not affect motility when LrhA is absent, based on motility in 0.3% agar for WT (*crl⁺*) cells or Δ*lrhA* cells (GSO1179) carrying an empty vector or overexpressing UhpU. Graph shows the average of three biological repeats, and error bars represent one SD. One-way ANOVA comparison was performed to calculate the significance of the change in motility (ns = not significant, ****=$p < 0.0001$). (**E**) Predicted base-pairing between *lrhA* and ArcZ, RprA or McaS. Numbering is from AUG of *lrhA* mRNA and +1 of indicated sRNAs. (**F**) Down regulation of *lrhA* by ArcZ and RprA but not McaS based on β-galactosidase assay detecting the levels of *lrhA-lacZ* translational fusions in response to ArcZ, RprA and McaS overexpression. For (**C**) and (**F**), graphs show the average of three biological repeats, and error bars represent one SD. One-way ANOVA comparison was performed to calculate the significance of the change in β-galactosidase activity (ns = not significant, ****=$p < 0.0001$).

The online version of this article includes the following figure supplement(s) for figure 3:

**Figure supplement 1.** Interactomes for σ²⁸-dependent sRNAs.

as cytosolic monomers (*Figure 4B*) and de-polymerized flagella (*Figure 4—figure supplement 1A*) with UhpU and MotR* overexpression and reduced levels with FliX or FliX-S overexpression. These differences are reflected in increased levels of the *fliC* mRNA with overexpression of UhpU, particularly in a *crl⁺* background, or MotR or MotR*, particularly at $OD_{600} \sim 0.2$ (*Figure 4C* and *Figure 4—figure supplement 1B*). In contrast, *fliC* mRNA levels decreased with FliX and FliX-S overexpression (*Figure 4C* and *Figure 4—figure supplement 1B*). In general, the impacts of the sRNAs on flagellin protein and *fliC* mRNA levels are consistent with the increased flagella number and/or motility upon UhpU or MotR overexpression and decreased flagella number upon FliX overexpression. Comparatively, the effects of MotR and MotR* on flagella number and *fliC* mRNA levels were stronger than the effects on the flagellin protein; possibly increases in flagellin levels are masked by the abundance of the protein.

We predicted base pairing between the three sRNAs and sequences overlapping the RIL-seq peaks internal to the *fliC* CDS (*Figure 4D*) and encompassing seed sequences suggested for the sRNAs (*Melamed et al., 2016*). To test for UhpU, MotR and FliX base pairing with these predicted sequences, we carried out in vitro footprinting with labeled fragments of the *fliC* mRNA (*Figure 4—figure supplement 2*). Upon cleavage with RNase III and lead, we observed changes in the regions predicted to be involved in base pairing (red brackets) that were dependent on the WT RNAs but not with derivatives carrying mutations in the regions predicted to be involved in base pairing. We also observed Hfq dependent changes (black bracket) in the region from ~+40 to+66 from the *fliC* AUG, which is enriched for ARN motif sequences (AAA, AAT, AAC, AAG, AAC), known to be important for mRNA binding to the distal face of Hfq binding (reviewed in *Updegrove et al., 2016*). Additionally, we noted that both MotR and the MotR-M1 mutant RNAs led to additional protection at another region (thin red bracket) and increased cleavage (red asterisks) at other positions and suggesting a second region of MotR base pairing with *fliC* as well as MotR-induced structure changes. In general, the differences in cleavage by RNase III (preference for double-stranded RNA) and lead (preference for single-stranded RNA), indicate the *fliC* sequence from ~+40 to~+170 is more structured than the surrounding regions. These differences in secondary structure could be the reasons for positive regulation by UhpU and MotR and negative regulation by FliX but also complicate analysis using standard reporter fusions with compensatory mutations.

## MotR and FliX modulate the S10 operon

Given that genes encoding ribosomal proteins were among the top MotR and FliX targets in the RIL-seq data sets and were not detected for many other sRNAs (*Supplementary file 1* and *Figure 3—figure supplement 1B*), we investigated MotR and FliX regulation of these genes. Several of the top interactions for MotR and FliX in the RIL-seq data mapped to the essential S10 operon, again within the CDSs (*Figure 5* and *Figure 3—figure supplement 1B*). The co-transcriptional regulation of the S10 operon has been studied extensively (*Zengel and Lindahl, 1996*; *Zengel et al., 2002*; *Zengel and Lindahl, 1992*). The leader sequence upstream of the first gene *rpsJ* encoding S10 is bound by the ribosomal protein L4, encoded by the third gene in the operon (*rplD*), causing transcription termination, thus modulating the levels of all the ribosomal proteins in the operon in response to the levels of unincorporated L4. L4 binding also has been shown to specifically inhibit translation of *rpsJ*, an effect that can be genetically distinguished from the L4 effect on transcription termination (*Freedman et al., 1987*).

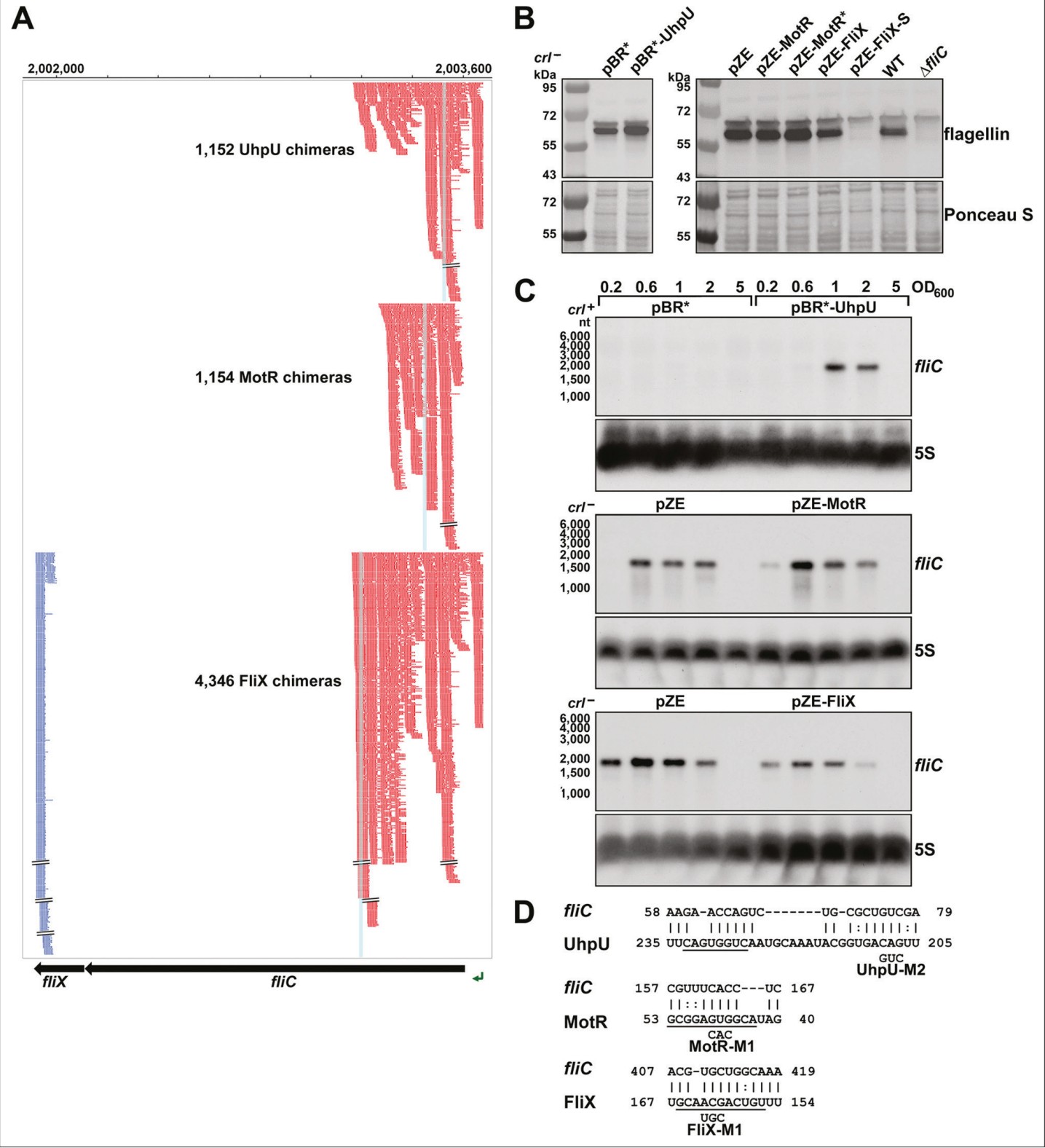

**Figure 4.** Multiple sRNAs regulate flagellin synthesis. (**A**) Browser image showing chimeras (red and blue) for UhpU, MotR, and FliX at the *fliCX* region. Data analyzed is from (RIL-seq experiment 1, *Melamed et al., 2020*). Red and blue lines indicate the RNA in the region is first or second RNA in the chimera, respectively. Blue highlighting indicates position of sRNA-*fliC* base pairing. (**B**) Immunoblot analysis showing UhpU and MotR overexpression leads to increased flagellin levels and FliX overexpression leads to reduced flagellin levels in the cytosol. Flagellin levels were determined by immunoblot analysis using α-FliC antibody. A sample from a Δ*fliC* strain was included as a control given the detection of a cross-reacting band slightly

*Figure 4 continued on next page*

*Figure 4 continued*

larger than flagellin. The Ponceau S-stained membrane serves as a loading control. Cells were grown with shaking at 180 rpm to OD$_{600}$ ~1.0, and cell fractions were separated by a series of centrifugation steps as detailed in Materials and Methods. (**C**) Northern blot analysis showing UhpU and MotR overexpression increases *fliC* mRNA levels and FliX overexpression reduces *fliC* levels across growth. The 5S RNA served as a loading control. The variation in *fliC* levels in the pBR* and pZE control samples is due to the different strain backgrounds (*crl*$^+$ versus *crl*) and the length of membrane exposure to film. (**D**) Predicted base-pairing between *fliC* and UhpU, MotR, or FliX. Seed sequences predicted by *Melamed et al., 2016* or by this study are underlined. Numbering is from AUG of *fliC* mRNA and +1 of indicated sRNAs.

The online version of this article includes the following figure supplement(s) for figure 4:

**Figure supplement 1.** Effects of UhpU, MotR* and FliX-S overexpression on flagellin and *fliC* mRNA levels.

**Figure supplement 2.** In vitro structural probing of interaction between UhpU, MotR, and FliX sRNAs with *fliC* mRNA.

To test for MotR regulation of *rpsJ* expression, we fused the S10 leader and part of the *rpsJ* CDS, including the position of the *rpsJ*-MotR chimeras (*Figure 5A*) and the region of predicted base-pairing (*Figure 5B*), to a GFP reporter (*Corcoran et al., 2012*; *Urban and Vogel, 2009*). MotR overexpression elevated the expression of the *rpsJ-gfp* fusion, and MotR* enhanced this effect (*Figure 5C*). Positive regulation of S10 expression by MotR and MotR* was similarly observed by immunoblot analysis of an N-terminal FLAG-tagged S10 protein encoded along with the S10 leader behind the heterologous promoter on a pBAD plasmid (*Figure 5D*). A mutation in the MotR seed sequence (MotR-M1 and MotR*-M1, *Figure 1—figure supplement 1A*) eliminated the up-regulation of the FLAG-tagged S10 (*Figure 5D*) and the MotR effect on flagella number (*Figure 5—figure supplement 1A*). To examine base pairing between MotR and the sequences internal to the *rpsJ* CDS, we carried out in vitro structure probing in the presence of Hfq (*Figure 5E* and *Figure 5—figure supplement 2A and B*). The RNase T1, RNase III and lead cleavage assays supported the position of the predicted base-pairing between MotR and *rpsJ* mRNA (red and blue brackets), indicating MotR binds to *rpsJ* at ~+150 nt in its CDS. Again, we detected Hfq binding (black bracket), here to the attenuator hairpin in the S10 leader sequence (*Figure 5—figure supplement 2B*), which has three ARN sequences (AGG, AGU and AAC). The M1 mutation eliminated binding in the predicted region of pairing but a complementary mutation in the corresponding region of *rpsJ* mRNA did not restore MotR binding (*Figure 5E*). We suggest that, as for the MotR target region of *fliC*, MotR binds to more than one site, the MotR target region of *rpsJ* is highly structured, and MotR and Hfq binding might all lead to conformational changes that compound the interpretation of the mutations.

Nevertheless, to further define the determinants needed for MotR-mediated up regulation, we generated a series of *rpsJ-gfp* fusions to include the leader and only the first seven amino acids of S10 removing the MotR base pairing site, to remove the S10 leader sequence, to remove stem D required for L4-mediated regulation, or to remove the attenuator hairpin stem E (*Figure 5—figure supplement 1B*). MotR-dependent regulation was eliminated for each of these constructs suggesting that S10 leader sequence is needed along with the MotR binding site internal to the *rpsJ* CDS for MotR-dependent regulation (*Figure 5—figure supplement 1B*). To test if Hfq binding to *rpsJ* is critical for the activation, we repeated the GFP reporter assay in an Hfq$^{Y25D}$ mutant defective for binding ARN sequences on the distal face of the protein (*Zhang et al., 2013b*). Supporting a role for Hfq, MotR, which is present at the same levels in the Hfq WT and Hfq$^{Y25D}$ mutant strains, no longer upregulates *rpsJ-gfp* in the distal face mutant background (*Figure 5—figure supplement 1C*). Collectively, our results are consistent with MotR base pairing internal to *rpsJ* affecting Hfq binding to the S10 leader sequence, which in turn results in increased *rpsJ* translation.

Based on the RIL-seq data, FliX interacts with multiple regions in the S10 operon mRNA, all internal to CDSs (*Figure 5F*). The predicted base-pairing regions (*Figure 5G*) align with the highest peaks of chimeras in the RIL-seq data and overlap with the seed sequence suggested for FliX (*Melamed et al., 2016*). We tested the effects of FliX on expression from this operon by constructing *gfp* fusions to regions of *rplC*, *rpsQ*, and *rpsS-rplV*. In all cases, overproduction of FliX or FliX-S led to a reduction in the expression of these fusions (*Figure 5H*). To test for a direct interaction between FliX-S and the *rpsS* mRNA, we again carried out structure probing (*Figure 5I* and *Figure 5—figure supplement 2C and D*). The regions that were changed in *rpsS* and FliX-S in the in vitro footprinting aligned with the predicted binding region between the two RNAs. Introduction of the M1 mutation (*Figure 1—figure supplement 1A*) eliminated FliX-S binding to the *rpsS* mRNA while introduction of a complementary mutation in the *rpsS* mRNA restored FliX-S-M1 binding (*Figure 5I*). We hypothesize that FliX

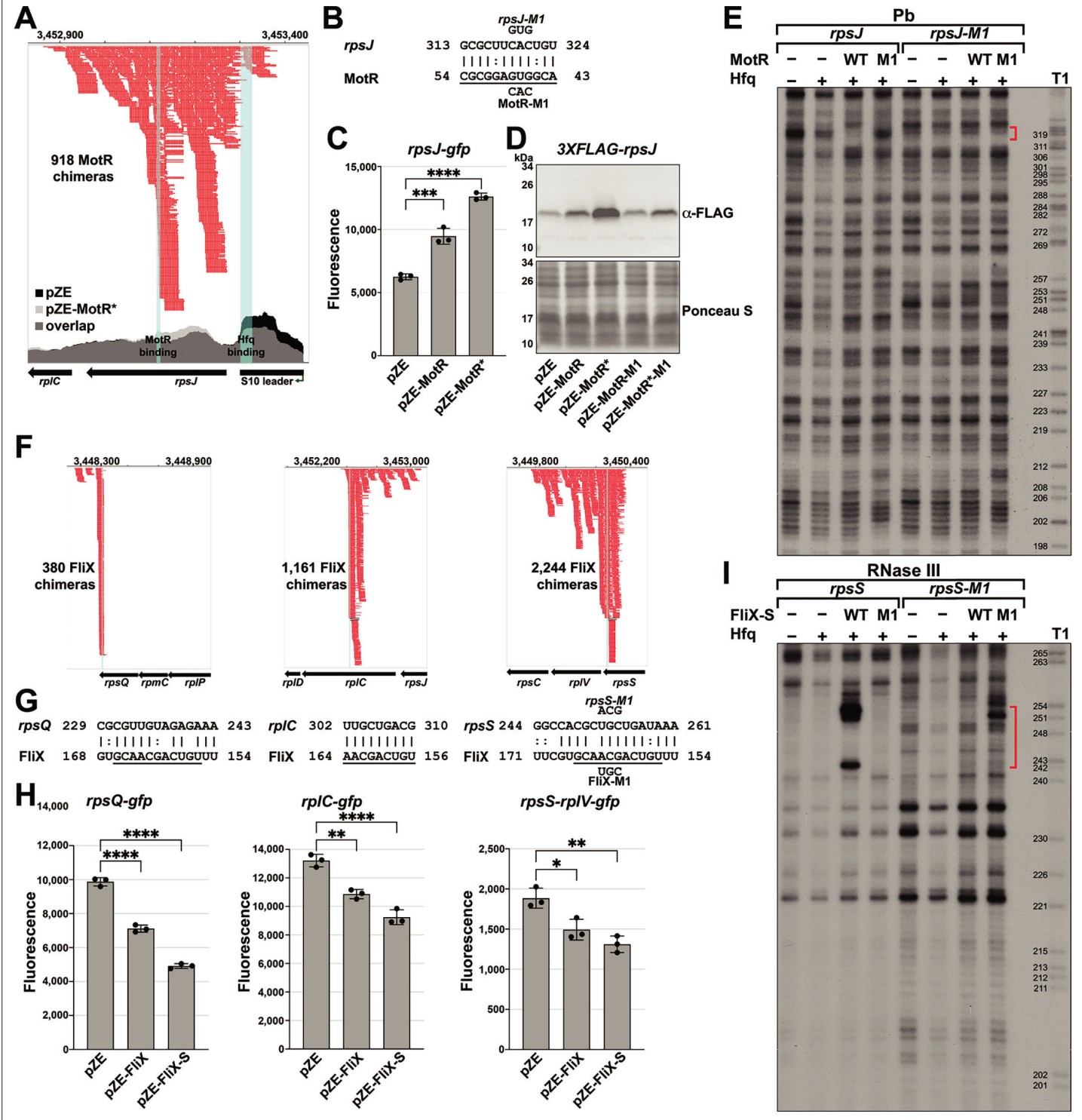

**Figure 5.** MotR and FliX base pair with the S10 mRNA leading to upregulation and downregulation, respectively. (**A**) Browser image showing MotR chimeras (in red) in S10 leader and *rpsJ* region. Data analyzed is from (RIL-seq experiment 1, *Melamed et al., 2020*). Coverage of the region in total RNA-seq libraries is shown for empty vector (pZE) and for pZE-MotR* overexpression (*Supplementary file 2*). The MotR and Hfq binding sites as detected in *Figure 5—figure supplement 2A and B* are highlighted in light blue. (**B**) Base-pairing between *rpsJ* and MotR with sequences of mutants assayed. Predicted MotR seed sequence is underlined. Numbering is from +1 of *rpsJ* mRNA and MotR sRNA. (**C**) MotR induces *rpsJ-gfp* reporter fusion based on reporter assays of *rpsJ-gfp* expressed from pXG10-SF with MotR or MotR* expressed from pZE. (**D**) MotR increases FLAG-tagged S10 levels. 3XFLAG-S10 was expressed from pBAD33 and MotR or MotR* was expressed from pZE. A mutation in MotR eliminates this regulation. 3XFLAG-S10 levels were determined by immunoblot analysis using α-FLAG antibody. The Ponceau S-stained membrane serves as a loading control. (**E**) Changes in

*Figure 5 continued on next page*

*Figure 5 continued*

RNase III-mediated cleavage of *rpsJ* due to MotR. [32]P-labeled *rpsJ* and *rpsJ*-M1 were treated with lead for 10 min with or without MotR and MotR-M1 and separated on a sequencing gel. Region protected by MotR binding, which overlaps the predicted base pairing sequence, is indicated by the red bracket. Numbering is from +1 of *rpsJ* mRNA. (**F**) Browser image showing FliX chimeras (in red) in the S10 operon. Highlighted in light blue are the base pairing regions between FliX and the S10 operon mRNA. Data analyzed is from (RIL-seq experiment 1, *Melamed et al., 2020*). (**G**) Base pairing between *rplC*, *rpsS*, *rpsQ*, and FliX with sequences of mutants assayed. FliX seed sequence predicted by *Melamed et al., 2016* is underlined. Numbering is from AUG of indicated CDS and +1 of FliX sRNA. (**H**) Test of FliX interactions with reporter assays of *rplC-gfp*, *rpsS-rplV-gfp*, and *rpsQ-gfp* expressed from pXG10-SF or pXG30-SF and FliX or FliX-S expressed from pZE. (**I**) Changes in RNase III-mediated cleavage of *rpsS* due to FliX-S. [32]P-labeled *rpsS* and *rpsS*-M1 were treated with RNase III for 1.5 min with or without FliX-S and FliX-S -M1 and separated on a sequencing gel. Region protected by FliX binding, which overlaps the predicted base pairing sequence, is indicated by the red bracket. Numbering is from AUG of *rpsS* CDS. For (**C**) and (**H**), the average of three independent measurements is shown. Error bars represent one SD. One-way ANOVA comparison was performed to calculate the significance of the change in GFP signal (ns = not significant, \*=p < 0.05, \*\*=p < 0.01, \*\*\*\*=p < 0.0001).

The online version of this article includes the following figure supplement(s) for figure 5:

**Figure supplement 1.** Effects of MotR mutants on flagella number and *rpsJ* expression.

**Figure supplement 2.** In vitro structural probing of interaction between MotR sRNA and *rpsJ* mRNA, and FliX sRNA and *rpsS* mRNA.

**Figure supplement 3.** In vivo effects of FliX and FliX-S overproduction on *rpsS* mRNA.

downregulation of the *rplC*, *rpsQ*, and *rpsS-rplV* fusions as well as the *fliC* mRNA is due to sRNA-directed mRNA degradation. Further experiments are needed to test this model, but in vivo primer extension assays carried out for RNA isolated from in mid-log phase cells (OD$_{600}$ ~0.6) showed an increase in 5′ ends in proximity to the binding site on the *rpsS* mRNA in FliX or FliX-S overexpressing strains (***Figure 5—figure supplement 3***).

## Increased S10 levels correlate with increased readthrough of flagellar operons

We wondered how the positive regulation of *rpsJ* by MotR might impact flagella synthesis. The S10 protein encoded by *rpsJ* has two roles in the cell. It is incorporated into the 30S ribosome subunit but also forms a transcription anti-termination complex with NusB (***Lüttgen et al., 2002***; ***Luo et al., 2008***; ***Baniulyte et al., 2017***). We evaluated the importance of each of the two S10 roles to flagella number by EM. First, we overexpressed a S10 mutant (S10Δloop) that is missing the ribosome binding loop but is still active in anti-termination (***Luo et al., 2008***) from an inducible plasmid and analyzed the number of flagella per cell. Cells carrying the S10Δloop plasmid had higher number of flagella like cells over-expressing MotR\* (***Figure 6A***). We noted that overexpression of wild type S10 from the plasmid used for overexpression of S10Δloop did not lead to an increase in flagella number (***Figure 6A***), although presumably MotR is normally increasing flagella number by impacting the levels of the WT protein. Possibly, only a specific concentration of S10 relative to other ribosome proteins increases the S10 role as an anti-terminator. Since *rpsJ* is essential and cannot be deleted, we also examined the effect of MotR\* overexpression in a Δ*nusB* strain that cannot form the S10-NusB anti-termination complex. In this background, the stimulatory effect of MotR\* on flagella number was eliminated (***Figure 6B***) as is also observed for S10Δloop overexpression in the Δ*nusB* background (***Figure 6—figure supplement 1A***). Based on these observations, we hypothesized that increased S10 levels upon MotR overexpression leads to increased anti-termination of some of the long flagella operons.

To directly test this anti-termination hypothesis, we carried out RT-qPCR analysis in WT and Δ*nusB* backgrounds to examine the effects of MotR and MotR\* overexpression on genes in the *motAB-cheAW* and *tar-tap-cheRBYZ* operons. For both operons, the mRNA levels of the tested genes were increased in WT upon MotR and MotR\* overexpression (***Figure 6C*** and ***Figure 6—figure supplement 1B***). This increase was not observed for the non-flagellar control gene *cadB*. While the levels of the flagellar mRNAs in Δ*nusB* background were lower than in the WT, MotR and MotR\* no longer induced these genes. Together these observations are consistent with the proposal that increased levels of non-ribosome associated S10 lead to increased levels of the S10-NusB anti-termination complex associated with RNA polymerase-σ$^{28}$ and increased anti-termination of the long operons encoding flagellar proteins. It is also conceivable that even a slight upregulation of the S10 operon, as well as the S6 operon, given a significant number of MotR-*rpsF* chimeras (***Supplementary file 1***), along with anti-termination of *rrn* operons, could lead to more active ribosomes, which are needed for flagellar

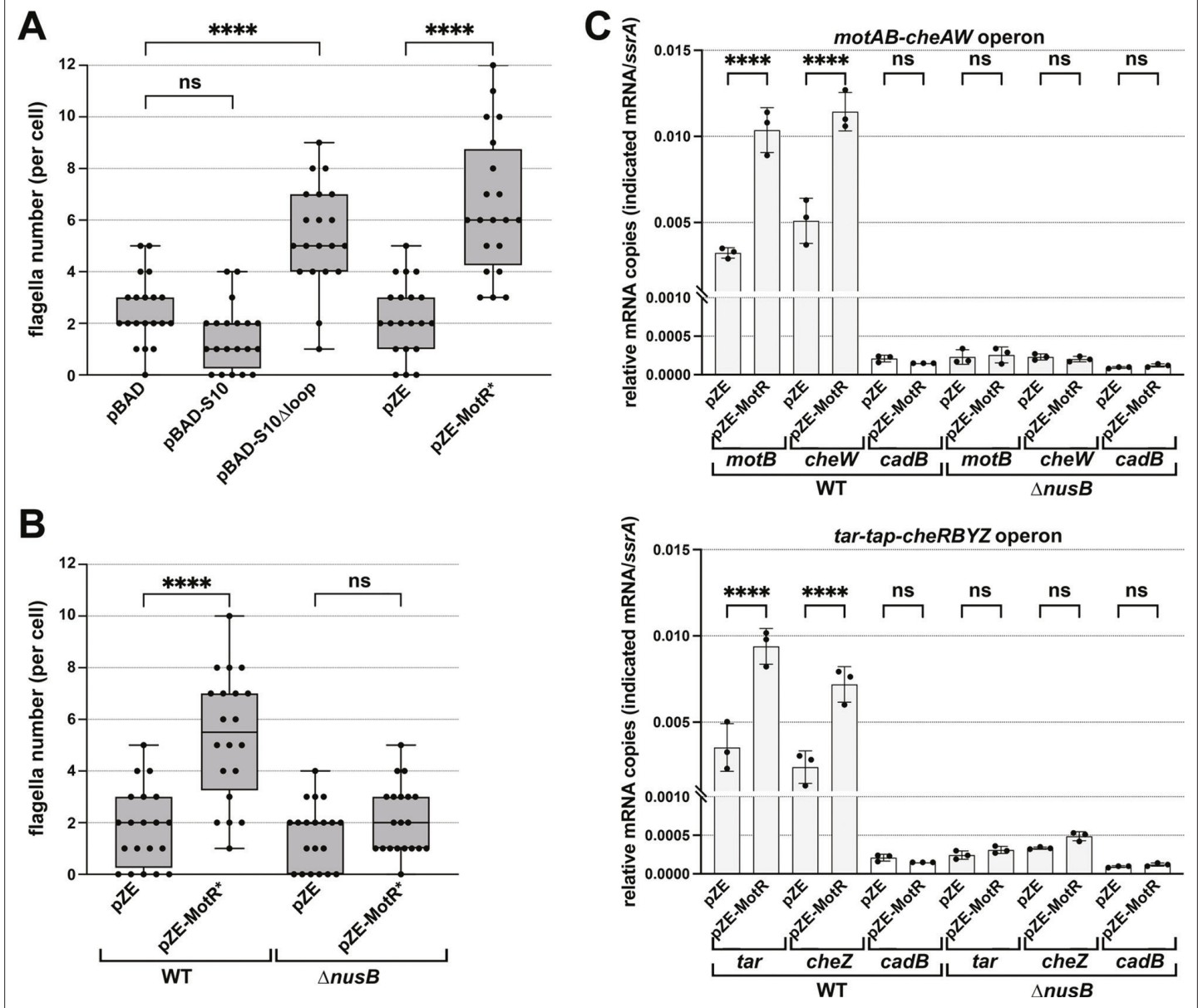

**Figure 6.** MotR overexpression leads to a *nusB*-dependent increase in expression from flagellar operons. (**A**) MotR* and S10Δloop overexpression increase the number of flagella. The number of flagella per cell detected by EM were counted for WT cells (GSO983) harboring the indicated plasmids. (**B**) MotR effect is eliminated in Δ*nusB* background. The number of flagella per cell detected by EM were counted for WT (GSO983) or Δ*nusB* cells (GSO1077) harboring the indicated plasmids. (**C**) MotR induces mRNA levels throughout the flagellar operons in WT background (GSO983) but not in Δ*nusB* background (GSO1077). MotR was expressed from pZE plasmid and the levels of *motB*, *cheW*, *tar*, *cheZ*, *ssrA* and *cadB* were monitored in comparison to their levels in the pZE control vector by RT-qPCR. *cadB* served as a non-flagellar gene control and *ssrA* served as a reference gene; the same *cadB* data is shown in both plots. Experiments were done in three biological replicates and one-way ANOVA comparison was performed to calculate the significance of the change in mRNA levels (ns = not significant, ****=p < 0.0001). For (**A**) and (**B**), flagella were counted for 20 cells (black dots), and a one-way ANOVA comparison was performed to calculate the significance of the change in flagella number (ns = not significant, ****=p < 0.0001). Box plot and error bars descriptions as in *Figure 2*. Each experiment was repeated three times, and one representative experiment is shown.

The online version of this article includes the following figure supplement(s) for figure 6:

**Figure supplement 1.** Effects of MotR* and S10Δloop overexpression are lost in Δ*nusB* background.

protein synthesis. On the other hand, a negative effect of FliX on ribosomal components, which could reduce the number of active ribosomes, would be consistent with the repressive role of this sRNA.

## MotR and FliX have opposing effects on the expression of middle and late flagella genes

In a parallel line of experimentation, we examined the impact of overexpressing MotR* and FliX on the transcriptome by RNA-seq analysis (*Supplementary file 2*). The transcripts whose levels increased most with MotR* overexpression compared to the vector control (*Figure 7A*) corresponded predominantly to late genes and, to a lesser extent, middle genes, of the flagellar regulon. Of the 332 genes whose expression increased significantly (FDR = 0.05) by MotR* overexpression, 40 are reduced significantly (FDR = 0.05) in a strain lacking σ$^{28}$ (Δ*fliA*) (*Fitzgerald et al., 2014*; *Figure 7— figure supplement 1A*). Additionally, the sequence motif found for the promoters of the transcription units for which expression increased the most (FDR = 0.05 and ≥2 fold) upon MotR* overproduction (*Figure 7—figure supplement 1B*) is nearly identical to a σ$^{28}$ recognition motif (*Fitzgerald et al., 2014*; *Shi et al., 2020*). In contrast, transcripts for flagellar genes were reduced by FliX overexpression (*Figure 7B*). Specifically, 28 of 149 genes for which the expression is reduced significantly (FDR = 0.05) are middle or late genes of the flagellar regulon (*Fitzgerald et al., 2014*). We note that we did not observe differential levels of the S10 operon transcript in the RNA-seq analysis upon FliX overexpression but did detect decreased levels of some transcripts encoding ribosomal proteins upon MotR* overexpression (*Figure 7B* and *Supplementary file 2*). However, the total RNA for the RNA-seq experiments was isolated from cells early in growth (OD$_{600}$ ~0.2).

The effects of MotR, MotR*, FliX, and FliX-S on flagella gene expression were further examined by monitoring fluorescence from *gfp* fused to the promoters of *flgB*, a representative Class 2 promoter, and *fliL*, a representative Class 2/3 promoter (*Zaslaver et al., 2006*). MotR and MotR* overexpression increased the activity of the two promoters, while FliX and FliX-S overexpression led to a reduction of their activity (*Figure 7C and D*, *Figure 7—figure supplement 2A and B*). The levels of C-terminally SPA-tagged FlgJ, also encoded by a Class 2 gene, similarly increased across growth upon MotR* overexpression, particularly early in growth, and decreased upon FliX-S overexpression (*Figure 7— figure supplement 2C and D*). The data suggest that in addition to modulating anti-termination and/ or ribosomal protein synthesis (*Figure 6*), MotR and FliX more broadly effect transcription initiation at flagellar genes though we do not know the mechanism. In general, these results are coherent with a positive effect of MotR on flagella synthesis and a negative effect of FliX.

## MotR increases and FliX decreases flagella synthesis

To examine the impact of chromosomally-encoded MotR and FliX on flagella synthesis and the flagellar regulon, we introduced the three-nucleotide M1 substitutions in the seed sequences of *motR* and *fliX* (MotR-M1 and FliX-M1, *Figure 1—figure supplement 1A*) at their endogenous chromosomal positions, avoiding the disruption of the nearby genes. MotR-M1 levels were comparable to WT MotR levels (*Figure 8—figure supplement 1A*). The prominent ~200 nt FliX band was reduced for FliX-M1, while other FliX processing products were affected less (*Figure 8—figure supplement 1B*).

We first examined the flagella number and motility for these strains. The *motR-M1* chromosomal mutation was associated with a moderate reduction in flagella number at two time points (OD$_{600}$ ~0.6 and 2.0) (*Figure 8A*), while slightly higher numbers of flagella were observed for the *fliX-M1* strain at the later time point (OD$_{600}$ ~2.0) (*Figure 8B*). In motility assays carried out as in *Figure 2*, we found reduced motility of the *motR-M1* strain compared to WT but no change was observed for the *fliX-M1* strain (*Figure 8—figure supplement 1C and D*). We also compared the motility of the *motR-M1* and *fliX-M1* strains to WT strains by mixing strains transformed with plasmids expressing either GFP or mCherry. WT strains expressing GFP were mixed with *motR-M1* or *fliX-M1* cells expressing mCherry or vice versa, and their motility was compared on 0.3% agar plates. For both combinations of WT and *motR-M1*, the fluorescent signal produced by the WT strain was more extensive than the fluorescent signal generated by *motR-M1* mutant outside of the site of inoculation (*Figure 8C*). Thus, in two independent assays, the *motR-M1* mutant exhibits reduced motility compared to the WT strain, while no significant difference was observed between WT and *fliX-M1* (*Figure 8D*).

We also assessed the effects of the chromosomal mutations on the *flgB-gfp* and *fliL-gfp* fusions (*Figure 8*) as well as on FlgJ-SPA and *fliC* mRNA levels (*Figure 8—figure supplement 1*). The *motR-M1*

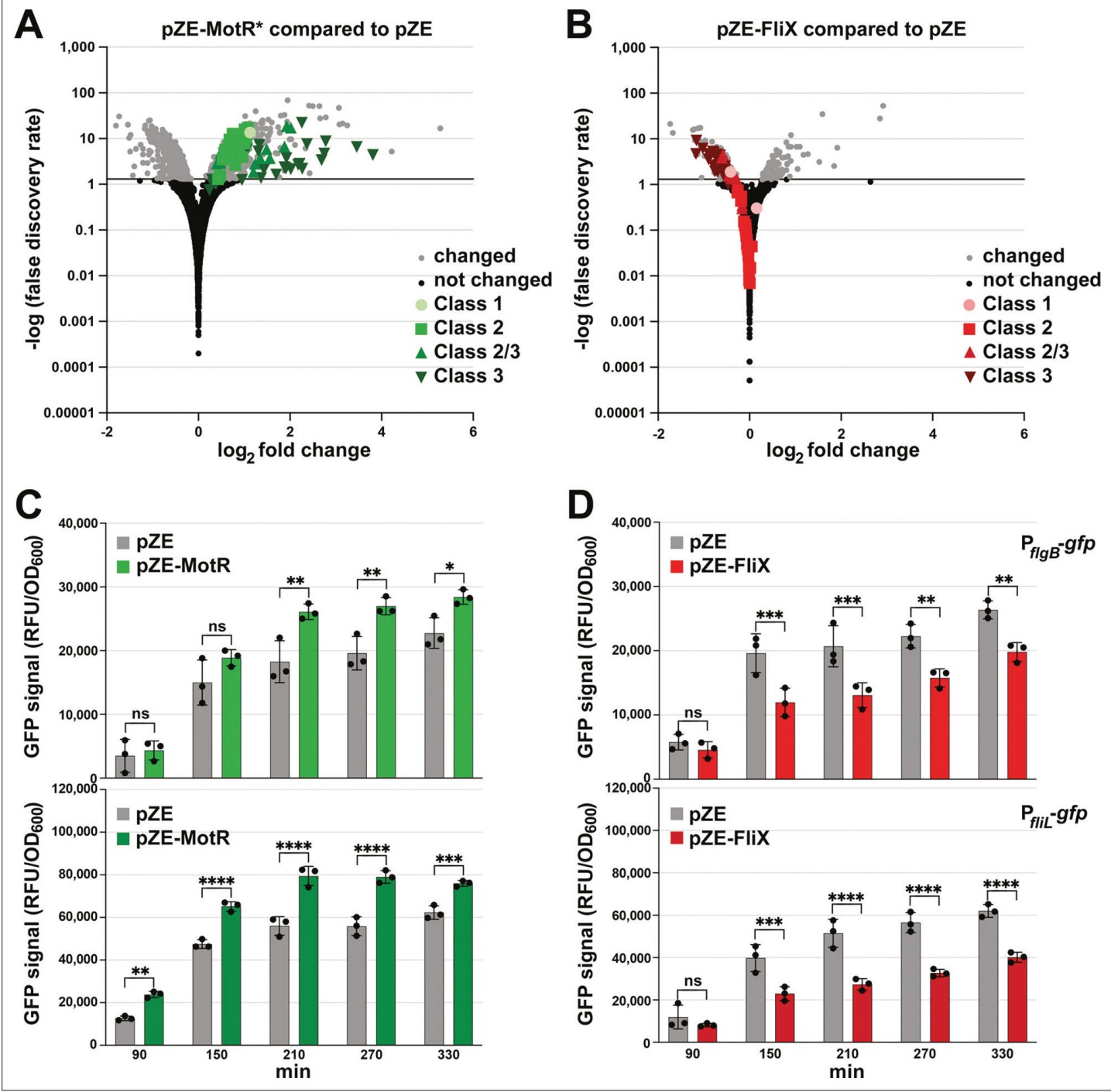

**Figure 7.** MotR and FliX overproduction leads to increased and decreased expression of flagellar genes, respectively. (**A**) MotR* induces flagellar genes. Green symbols represent flagellar regulon genes as indicated on the graph. (**B**) FliX reduces flagellar genes. Red symbols represent flagellar regulon genes as indicated on the graph. In (**A**) and (**B**), differential expression analysis was conducted with DESeq2. The threshold for differentially expressed transcripts was set to adjusted value of $P<0.05$. (**C**) MotR overexpression increases the activity of *gfp* fusions to $P_{flgB}$ and $P_{fliL}$. (**D**) FliX overexpression decreases the activity of *gfp* fusions to $P_{flgB}$ and $P_{fliL}$. In (**C**) and (**D**), the promoter activities were monitored for 330 min by measuring the GFP signal and dividing it with the culture $OD_{600nm}$. For (**A**) and (**B**), WT (GSO983) harboring the control vector pZE or the MotR* or the FliX expression plasmid were grown to $OD_{600}$ ~0.2; total RNA was extracted and used for the construction of cDNA libraries, which were analyzed as described in Materials and methods. For (**C**) and (**D**), three biological repeats are shown in the graph. One-way ANOVA comparison was performed to calculate the significance of the change in GFP signal (ns = not significant, *=p < 0.05, **=p < 0.01, ***=p < 0.001, ****=p < 0.0001). The experiments presented in (**C**) and

*Figure 7 continued on next page*

Figure 7 continued

Figure 7—figure supplement 2B, and in (D) and Figure 7—figure supplement 2A, were carried out on same day, respectively, and the same pZE samples are shown.

The online version of this article includes the following figure supplement(s) for figure 7:

Figure supplement 1. Overlap in MotR* overexpression profile with σ²⁸ regulon.

Figure supplement 2. Effects of MotR* and FliX-S overexpression on P_{flgB}-gfp, P_{fliL}-gfp and FlgJ-SPA expression.

mutant showed reduced activity of the two promoters (Figure 8E), as expected given the increased activity of the promoters that was observed upon MotR overexpression (Figure 7C). The fliX-M1 mutant showed similar activity of the two promoters in comparison to WT (Figure 8F). In western and northern analyses of the motR-M1 strain compared to its parental WT, a delayed initiation of FlgJ-SPA and fliC mRNA synthesis, respectively, was observed in the mutant (Figure 8—figure supplement 1E and G). In contrast, FlgJ-SPA and fliC mRNA levels increased in the fliX-M1 strain compared to the parental WT strain (Figure 8—figure supplement 1F and H).

While negative effects of the motR-M1 mutation on flagella number, motility, and flagellar gene expression were observed in all assays, positive effects of the fliX-M1 mutation were only detected for flagella number, FlgJ-SPA protein, and fliC mRNA levels. However, for both sRNAs the mutation phenotype is opposite that of the overexpression phenotype. Collectively these observations indicate that MotR, expressed earlier in growth, increases flagella synthesis by positively regulating the middle and the late genes, while FliX, whose levels peak later, decreases flagella synthesis by downregulating the flagellar regulon. Thus, MotR and FliX, along with UhpU, add another layer of regulation to the flagellar regulon (Figure 8G).

## Discussion

In this study, we describe four E. coli sRNAs whose expression is dependent on σ²⁸. We found three of these sRNAs affect flagella number and bacterial motility. Although previous studies showed that base pairing sRNAs act on the flhDC mRNA (Thomason et al., 2012; De Lay and Gottesman, 2012; Lejars et al., 2022), our results revealed that the effect of sRNAs on flagellar synthesis is far more pervasive. Intriguingly, two of the σ²⁸-dependent sRNAs show opposite effects. MotR, expressed earlier in growth, increases expression of flagellar and ribosomal proteins along with flagella number, while FliX, expressed later in growth, decreases expression of the proteins and flagella number. Thus, the two sRNAs, respectively, might be considered an accelerator and a decelerator for flagellar synthesis.

### Non-canonical mechanisms of sRNA action

Most commonly, sRNAs base pair with the 5′ UTRs of mRNA targets or at the very beginning of the CDS, primarily affecting ribosome binding or mRNA stability. However, MotR and FliX bind in the middle or even close to the ends of their target CDSs in the fliC gene and S10 operon. For both fliC and the S10 operon, the consequences of MotR and FliX overexpression are different. MotR leads to higher levels of fliC and the S10 protein, whereas FliX leads to lower levels of fliC and three genes in the S10 operon. We suggest that the positive and negative regulatory effects of MotR and FliX, respectively, occur by the same mechanisms on the fliC and S10 transcripts, with MotR changing the conformation of the RNAs and FliX leading to increased cleavage. However, these suggested mechanisms needed to be investigated further in future experiments. It is also noteworthy that, based on RIL-seq data, more examples of CDS internal interactions remain to be characterized.

Given that our study made extensive use of RIL-seq data, it provides an opportunity to evaluate these data. While RIL-seq provides a comprehensive map of RNA-RNA interactions that take place on Hfq under a specific condition, some caution about the interpretation is warranted as the interactions represent multiple types of relationships between two RNAs. As was found by a recent study (Faigenbaum-Romm et al., 2020), we suggest that if an interaction is highly abundant and discovered under multiple conditions, the sRNA is more likely to have a regulatory impact on the target mRNA though the mechanisms may be unknown. We noticed that the spread of the RIL-seq signal varies significantly between targets. One possible explanation for multiple peaks and a broad distribution is more than one base pairing site for the sRNA on the mRNA, but this hypothesis requires further

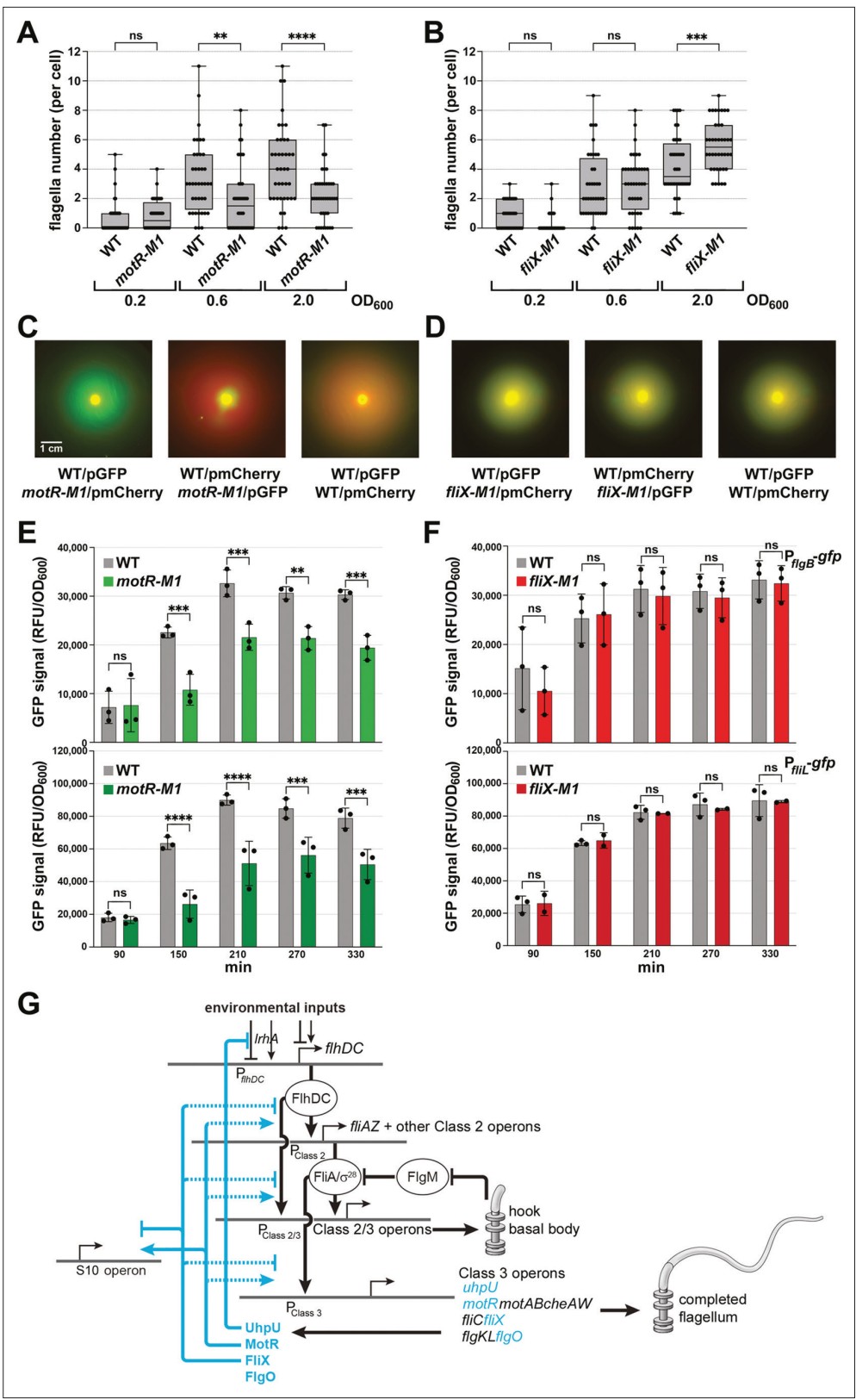

**Figure 8.** Complex regulatory network of sRNAs controlling flagella synthesis. (**A**) Reduction in flagella number in *motR-M1* mutant. (**B**) Increase in flagella number in *fliX-M1* mutant. (**C**) Reduced motility in *motR-M1* mutant (GSO1087) based on a competition assay with its corresponding WT (GSO1088). (**D**) No difference in motility in *fliX-M1* mutant (GSO1076) based on a competition assay with its corresponding WT (GSO983). (**E**) Reduction

*Figure 8 continued on next page*

*Figure 8 continued*

in P$_{flgB}$-*gfp* and P$_{fliL}$-*gfp* expression in *motR-M1* mutant (GSO1087) background compared to WT background (GSO1088). (**F**) No difference in P$_{flgB}$-*gfp* and P$_{fliL}$-*gfp* expression in *fliX-M1* mutant (GSO1076) background compared to WT background (GSO983). (**G**) σ$^{28}$-dependent sRNAs control flagella synthesis at different levels. UhpU activates the flagellar regulon by repressing a regulator of *flhDC*. MotR and FliX, respectively, activate and repress middle and the late gene expression (dotted line indicates exact mechanism is not known, although we document base pairing with the *fliC* mRNA). MotR and FliX also connect ribosome and flagella synthesis by regulating genes in the S10 operon (solid line indicates documented base pairing with this mRNA). In (**A**) and (**B**), the number of flagella per cell detected by EM were counted for 40 cells (black dots) for the *motR-M1* (GSO1087) and its corresponding WT (GSO1088), and for *fliX-M1* (GSO1076) and its corresponding WT (GSO983), strains at three points in growth (OD$_{600}$ ~0.2, OD$_{600}$ ~0.6, and OD$_{600}$ ~2.0). A one-way ANOVA comparison was performed to calculate the significance of the change in flagella number (ns = not significant, **=p < 0.01, ***=p < 0.001, ****=p < 0.0001). Each experiment was repeated three times and one representative experiment is shown. Box plot and error bars descriptions as in *Figure 2*. For (**C**) and (**D**), WT or the corresponding mutant, expressed either green fluorescent signal or red fluorescent signal by carrying pCON1.proC-GFP or pCON1.proC-mCherry plasmid, respectively. In the left images, WT cells expressing GFP were mixed with mutant cells expressing mCherry; in the middle images, WT cells expressing mCherry were mixed with mutant cells expressing GFP; in the right images, WT cells expressing GFP were mixed with WT cells expressing mCherry. The indicated mixed cultures were spotted on a soft agar (0.3%) plate, incubated at 30 °C, and imaged after 18 hr. The scale given in (**C**) is the same for all motility plates. For (**E**) and (**F**), three biological repeats are shown in the graph (except for P$_{fliL}$-*gfp* in *fliX-M1*, for which two repeats are shown). One-way ANOVA comparison was performed to calculate the significance of the change in GFP signal (ns = not significant, **=p < 0.01, ***=p < 0.001, ****=p < 0.0001).

The online version of this article includes the following figure supplement(s) for figure 8:

**Figure supplement 1.** Effects of chromosomal *motR-M1* and *fliX-M1* mutations.

**Figure supplement 2.** Conservation of σ$^{28}$-dependent sRNAs.

---

testing. We predict additional studies of sRNA-target pairs with different types of RIL-seq signals will give further insights into the mechanisms and outcomes of base pairing.

The most studied and conserved sRNA-binding protein in gram-negative bacteria is Hfq. However, there are other sRNA-binding proteins (reviewed in *Melamed, 2020*). Among these is ProQ, which was shown to have overlapping, complementary, and competing roles with Hfq in *E. coli* (*Melamed et al., 2020*). Interestingly, ProQ was found to affect motility and chemotaxis in *S. enterica* (*Westermann et al., 2019*). In the absence of ProQ, the target sets for the σ$^{28}$-dependent sRNAs on Hfq were changed significantly in *E. coli* (Table S5 in *Melamed et al., 2020*) suggesting that competition between Hfq and ProQ for binding RNAs likely also influences this regulatory circuit. In this context, it is worth noting that FlgO, the fourth σ$^{28}$-dependent sRNA, which originates from the 3′UTR of the *flgL* and strongly binds Hfq (*Melamed et al., 2020*), does not have many targets. Possibly, FlgO has a role in titrating Hfq from other sRNAs or proteins, or in the recruitment of other proteins to a complex with Hfq. Interestingly, while the overall sequence of *flgO* is conserved in other bacterial species (*Figure 8—figure supplement 2*), the nucleotides in one of the single stranded loops (*Figure 1—figure supplement 1C*) differ in *S. typhimurium*, possibly suggesting distinct regulatory mechanisms in different bacteria.

## Conservation of σ$^{28}$-dependent sRNAs

We were surprised to find so many σ$^{28}$-dependent Hfq-binding sRNAs and wondered about their phylogenic distribution. The σ$^{28}$-dependent sRNAs studied here are conserved among some of the Enterobacteriaceae (*Figure 8—figure supplement 2*) and thus may play a role in pathogenicity. Two studies describing the application of RIL-seq to *S. enterica* and Enteropathogenic *E. coli* Hfq were recently published (*Pearl Mizrahi et al., 2021*; *Matera et al., 2022*). The RIL-seq analyses were carried out for cells grown under conditions that do not favor flagellar gene expression, but UhpU, MotR, FliX, and FlgO were detected, confirming their synthesis and association with Hfq in pathogenic bacteria. Previous work assessing the conservation of the *motR* and *uhpU* promoters showed that, while the *motR* promoter is well conserved across proteobacteria species, the *uhpU* promoter was not, implying different evolutionary pressures (*Fitzgerald et al., 2018*). Interestingly, however, a sRNA named RsaG, which originates from the 3′ UTR of *uhpT* and also is induced by glucose-6-phosphate, was found in the Gram-positive bacterium, *Staphylococcus aureus* (*Bronesky et al.,*

*2019*). Although there is no sequence similarity between UhpU and RsaG, and RsaG has not been reported to regulate flagella synthesis, the independent evolution of regulatory sRNAs at the 3′ UTRs of *uhpT* in two disparate bacterial species is intriguing. RsaG was found to regulate redox homeostasis and to adjust metabolism to changing environmental conditions (*Desgranges et al., 2022*). While we focused on the UhpU role in the flagellar regulon and in controlling motility, the sRNA has many targets that are part of different metabolic pathways and redox homeostasis (*Supplementary file 1*), hinting at parallels between the two sRNAs.

It is likely that several other sRNA regulators of the flagellar regulon remain to be characterized. In *S. enterica*, a leader RNA originating from the *mgtCBR* virulence operon was shown to affect the synthesis of one of the two flagellin genes that exist in this bacterium, impacting virulence and motility (*Choi et al., 2017*). In neonatal meningitis-causing *E. coli*, a sRNA missing from the *E. coli* MG1655 strain used in our study, was shown to reduce *fliC* mRNA levels (*Sun et al., 2022*). Additionally, a very recent study of the *Campylobacter jejuni* FlmE and FlmR sRNAs showed that these two sRNAs have opposite effects on flagellar gene expression (*König et al., 2023*), resembling the opposing effects of MotR and FliX in *E. coli.*

## Roles of σ²⁸-dependent sRNAs

The UhpU RIL-seq target set includes many flagellar regulon genes and some transcription regulators of the flagellar regulon, such as LrhA (*Lehnen et al., 2002*), hinting at a mechanism by which UhpU can affect flagella number and bacterial motility. However, since UhpU can also be derived from the *uhpT* mRNA (*Figure 1—figure supplement 1B*) and is predicted to have many targets that participate in carbon and nutrient metabolism (*Supplementary file 1*), we suggest this sRNA may play a broader role in linking carbon metabolism with flagella synthesis and motility.

MotR and FliX each have more limited target sets in the RIL-seq data but may comprise a unique regulatory toggle. While the transcription of the two sRNAs is dependent on the same sigma factor and they base pair in the CDS of targets in the same operons, base pairing results in opposing regulation. MotR, which is transcribed from within *flhC* at the top of the flagellar regulatory cascade, reaches its highest levels earlier than FliX and increases the flagella synthesis. In contrast, FliX, which is cleaved from the mRNA required to make the last protein needed to complete the flagellum, reaches its highest levels later in growth and appears to decrease flagella synthesis.

It is not yet clear how MotR and FliX base pairing with only a few targets can have pervasive effects on flagellar gene expression and flagella number, but we suggest multiple mechanisms may be involved. One possibility is that the levels of flagellin encoded by *fliC*, up and down regulated by MotR and FliX, respectively, could be part of an autoregulatory loop that impacts the transcription of *flhDC* or other middle or late flagellar gene promoters. The increased and decreased levels of ribosomal proteins brought about by MotR and FliX regulation of the S10 operon also could impact the levels of available ribosomes, where even slight changes could have consequences given the high ribosome cost of flagella synthesis. Finally, we hypothesize that elevated levels of the S10 protein, due to the regulation by MotR, could, in conjunction with NusB, lead to increased anti-termination of long flagellar operons.

Based on our hypothesis that the MotR-mediated increase in S10 levels leads to increased anti-termination, we speculate that MotR activation of S10 expression could serve an autoregulatory role. Early in growth, transcription initiating from the σ²⁸-dependent promoter in *flhC* terminates at the 5′ of *motA* generating MotR. As MotR levels increase, there is a concomitant increase in S10 levels, which could promote readthrough of the *motR* terminator leading to decreased MotR levels and increased full-length *motRAB-cheAW* mRNA. The proposed FliX-directed cleavage of the *fliC* mRNA could have a similar negative feedback role, the cleavage would lead to less full-length *fliC* mRNA resulting in less FliX.

In general, the σ²⁸-dependent sRNAs add a new layer of regulation to the flagellar regulon and reinforce the conclusion that flagella synthesis is exquisitely regulated. The regulon will continue to serve as a model of a temporal and environmentally controlled regulatory network with contributions from both transcription factors and regulatory RNAs.

# Materials and methods

**Key resources table**

| Reagent type (species) or resource | Designation | Source or reference | Identifiers | Additional information |
|---|---|---|---|---|
| Chemical compound, drug | TRIzol Reagent | Thermo Fisher Scientific | Cat#15596018 | |
| Chemical compound, drug | 212–300 µm glass beads | Sigma-Aldrich | Cat#G1277 | |
| Chemical compound, drug | Protein A-Sepharose beads CL-4B | GE Healthcare | Cat#17-0780-01 | |
| Chemical compound, drug | Ureagel-8 | National Diagnostics | Cat#EC-838 | |
| Chemical compound, drug | Ureagel Complete | National Diagnostics | Cat#EC-841 | |
| Chemical compound, drug | NuSieve 3:1 Agarose | Lonza | Cat#50090 | |
| Chemical compound, drug | 37% Formaldehyde | Fisher Scientific | Cat#BP531-500 | |
| Commercial assay or kit | RiboRuler High Range RNA Ladder | Thermo Fisher Scientific | Cat#SM1821 | |
| Commercial assay or kit | RiboRuler Low Range RNA Ladder | Thermo Fisher Scientific | Cat#SM1831 | |
| Commercial assay or kit | Zeta-Probe GT membrane | Bio-Rad | Cat#1620159 | |
| Chemical compound, drug | ULTRAhyb-Oligo Hybridization Buffer | New England Biolabs | Cat#AM8663 | |
| Chemical compound, drug | [γ-$^{32}$P] ATP | PerkinElmer | Cat#NEG035C010MC | |
| Commercial assay or kit | T4 Polynucleotide Kinase | New England Biolabs | Cat#M0201L | |
| Commercial assay or kit | Illustra MicroSpin G-50 Columns | GE Healthcare | Cat#27533001 | |
| Commercial assay or kit | Mini-PROTEAN TGX Gels | Bio-Rad | Cat#456–1086 | polyacrylamide SDS gel |
| Commercial assay or kit | Nitrocellulose Membrane | Thermo Fisher Scientific | Cat#LC2000 | |
| Chemical compound, drug | RNase III | Fisher Scientific | Cat#AM2290 | |
| Commercial assay or kit | QuikChange Lightning Site-Directed Mutagenesis Kit | Agilent | Cat#210519 | |
| Commercial assay or kit | Amersham ECL Western Blotting Detection Kit | GE Healthcare | Cat#RPN2108 | |
| Commercial assay or kit | MEGAshortscript T7 High Yield Transcription Kit | Thermo Fisher Scientific | Cat#AM1354 | |
| Commercial assay or kit | Ambion RNase T1 Kit | Thermo Fisher Scientific | Cat#AM2283 | |
| Commercial assay or kit | iTaq Univer SYBR Green mix | Bio-Rad | Cat#1725124 | |
| Antibody | Mouse monoclonal ANTI-FLAG M2-Peroxidase | Sigma-Aldrich | Cat#A8592 | 1:1,000 |
| Antibody | Rabbit polyclonal anti-flagellin | Abcam | Cat#ab93713 | 1:5,000 |

*Continued on next page*

*Continued*

| Reagent type (species) or resource | Designation | Source or reference | Identifiers | Additional information |
|---|---|---|---|---|
| Sequence-based reagent | Primers, probes and DNA fragments | this study | **Supplementary file 3** | For requests, see "Data and Materials Availability" section |
| Strain, strain background (NM400) | NM400 (MG1655, *mini-λ, cmR, ts*) | A gift from Nadim Majdalani (S. Gottesman lab) | NM400 | |
| Strain, strain background (MG1655 (crl⁻)) | SMS001 (MG1655 (*crl⁻*)) | lab stock | GSO983 | |
| Strain, strain background (MG1655 (crl⁺)) | SMS046 (MG1655 (*crl⁺*)) | lab stock | GSO982 | |
| Strain, strain background (BW25113) | JW0406 (BW25113 Δ*nusB::kan*) | **Baba et al., 2006** | JW0406 | |
| Strain, strain background (BW25113) | JW2284 (BW25113 Δ*lrhA::kan*) | **Baba et al., 2006** | JW2284 | |
| Strain, strain background (MC4100) | SMS078 (MC4100; *hfq⁺*) | **Zhang et al., 2013b** | GSO614 | |
| Strain, strain background (AMD061) | SMP284 (AMD061 (MG1655 Δ*thyA* +pKD46)) | **Stringer et al., 2012** | SMP284 | |
| Strain, strain background (PM1205) | PM1205 (MG1655 *mal::lacIq, ΔaraBAD, lacI'':: PBAD-cat-sacB:lacZ, mini λ tetR*) | **Mandin and Gottesman, 2009** | PM1205 | |
| Strain, strain background (MC4100) | SMS079 (MC4100 *hfq-Y25D*) | this study | GSO1110 | For requests, see "Data and Materials Availability" section |
| Strain, strain background (NM400) | SMS007 (NM400 Δ*fliA::kan*) | this study | GSO1111 | For requests, see "Data and Materials Availability" section |
| Strain, strain background (MG1655 (crl⁻)) | SMS012 (MG1655 (*crl⁻*) Δ*fliA::kan*) | this study | GSO1068 | For requests, see "Data and Materials Availability" section |
| Strain, strain background (NM400) | SMS031 (NM400 Δ*fliCX::kan*) | this study | GSO1071 | For requests, see "Data and Materials Availability" section |
| Strain, strain background (MG1655 (crl⁻)) | SMS033 (MG1655 (*crl⁻*) Δ*fliCX::kan*) | this study | GSO1072 | For requests, see "Data and Materials Availability" section |
| Strain, strain background (MG1655 (crl⁻)) | SMS035 (MG1655 (*crl⁻*) Δ*fliCX*) | this study | GSO1073 | Δ*fliC*, for requests, see "Data and Materials Availability" section |
| Strain, strain background (NM400) | SM215 (NM400 *fliX-M1::kan*) | this study | GSO1074 | For requests, see "Data and Materials Availability" section |
| Strain, strain background (MG1655 (crl⁻)) | SMS249 (MG1655 (*crl⁻*) *fliX-M1::kan*) | this study | GSO1075 | For requests, see "Data and Materials Availability" section |
| Strain, strain background (MG1655 (crl⁻)) | SMS251 (MG1655 (*crl⁻*) *fliX-M1*) | this study | GSO1076 | *fliX-M1*, for requests, see "Data and Materials Availability" section |
| Strain, strain background (MG1655 (crl⁻)) | SMS044 (MG1655 (*crl⁻*) Δ*nusB::kan*) | this study | GSO1077 | Δ*nusB*, for requests, see "Data and Materials Availability" section |

*Continued on next page*

*Continued*

| Reagent type (species) or resource | Designation | Source or reference | Identifiers | Additional information |
|---|---|---|---|---|
| Strain, strain background (MC4100) | SMS216 (NM400 *flgJ-SPA::kan*) | this study | GSO1078 | For requests, see "Data and Materials Availability" section |
| Strain, strain background (MG1655 (crl⁻)) | SMS221 (MG1655 (*crl*) *flgJ-SPA::kan*) | this study | GSO1080 | For requests, see "Data and Materials Availability" section |
| Strain, strain background (MG1655 (crl⁻)) | SMS229 (MG1655 (*crl*) Δ*thyA flgJ-SPA::kan*) | this study | GSO1081 | For requests, see "Data and Materials Availability" section |
| Strain, strain background (MG1655 (crl⁻)) | SMS224 (MG1655 (*crl*) Δ*thyA motR-M1 flgJ-SPA::kan*) | this study | GSO1082 | For requests, see "Data and Materials Availability" section |
| Strain, strain background (MG1655 (crl⁻)) | SMS230 (MG1655 (*crl*) *fliX-M1 flgJ-SPA::kan*) | this study | GSO1083 | For requests, see "Data and Materials Availability" section |
| Strain, strain background (MG1655 (crl⁻)) | SMS202 (MG1655 (*crl*) Δ*thyA* +pKD46) | this study | GSO1085 | For requests, see "Data and Materials Availability" section |
| Strain, strain background (MG1655 (crl⁻)) | SMS209 (MG1655 (*crl*) *motR::thyA* +pKD46) | this study | GSO1086 | For requests, see "Data and Materials Availability" section |
| Strain, strain background (MG1655 (crl⁻)) | SMS210 (MG1655 (*crl*) Δ*thyA motR-M1*) | this study | GSO1087 | *motR-M1*, for requests, see "Data and Materials Availability" section |
| Strain, strain background (MG1655 (crl⁻)) | SMS213 (MG1655 (*crl*) Δ*thyA*) | this study | GSO1088 | *motR-M1* corresponding WT, for requests, see "Data and Materials Availability" section |
| Strain, strain background (MG1655 (crl⁺)) | AS003 (MG1655 (*crl⁺*) Δ*lrhA::kan*) | this study | GSO1178 | For requests, see "Data and Materials Availability" section |
| Strain, strain background (MG1655 (crl⁺)) | AS004 (MG1655 (*crl⁺*) Δ*lrhA*) | this study | GSO1179 | Δ*lrhA*, for requests, see "Data and Materials Availability" section |
| Strain, strain background (PM1205) | SMS021 (PM1205 *lrhA:lacZ*) | this study | GSO1180 | *lrhA-lacZ fusion*, for requests, see "Data and Materials Availability" section |
| Strain, strain background (PM1205) | SMS050 (PM1205 *lrhA.m1:lacZ*) | this study | GSO1181 | *lrhA-M1-lacZ*, for requests, see "Data and Materials Availability" section |
| Recombinant DNA reagent (plasmid) | SMP269 (NEB5α+pKD4) | *Datsenko and Wanner, 2000* | | pKD4 |
| Recombinant DNA reagent (plasmid) | SMP046 (TOP10 +pCP20) | *Cherepanov and Wackernagel, 1995* | | pCP20 |
| Recombinant DNA reagent (plasmid) | SMP284 (MG1655 (*crl*)+pKD46) | *Datsenko and Wanner, 2000* | | pKD46 |
| Recombinant DNA reagent (plasmid) | SMP045 (NEB5α+pJL148) | *Zeghouf et al., 2004* | | pJL148 |
| Recombinant DNA reagent (plasmid) | SMP043 (MG1655 (*crl*)+pBR*) | *Guo et al., 2014* | | pBR* |
| Recombinant DNA reagent (plasmid) | SMP006 (MG1655 (*crl*)+pZE12 luc) | *Lutz and Bujard, 1997* | | pZE12-luc |
| Recombinant DNA reagent (plasmid) | SMP004 (MG1655 (*crl*)+pZE (pJV300)) | *Urban and Vogel, 2007* | | pZE |

*Continued on next page*

*Continued*

| Reagent type (species) or resource | Designation | Source or reference | Identifiers | Additional information |
|---|---|---|---|---|
| Recombinant DNA reagent (plasmid) | SMP001 (MG1655 (crl⁻)+pXG0) | *Urban and Vogel, 2007* | | pXG0 |
| Recombinant DNA reagent (plasmid) | SMP002 (MG1655 (crl⁻)+pXG10 SF) | *Corcoran et al., 2012* | | pXG10-SF |
| Recombinant DNA reagent (plasmid) | SMP002 (MG1655 (crl⁻) pXG30-SF) | *Corcoran et al., 2012* | | pXG30-SF |
| Recombinant DNA reagent (plasmid) | SMP322 (NEB5α+pCON1.proC-GFP) | *Cooper et al., 2017* | | pCON1.proC-GFP |
| Recombinant DNA reagent (plasmid) | SMP323 (NEB5α+pCON1.proC-mCherry) | *Cooper et al., 2017* | | pCON1.proC-mCherry |
| Recombinant DNA reagent (plasmid) | SMP135 (MG1655 (crl⁻)+pBAD24) | *Guzman et al., 1995* | | pBAD24 |
| Recombinant DNA reagent (plasmid) | SMP164 (N9739 +pBAD nusE / pBAD-S10) | *Luo et al., 2008* | | pBAD-S10 |
| Recombinant DNA reagent (plasmid) | SMP165 (N9739 +pBAD-nusEΔloop / pBAD-S10Δloop) | *Luo et al., 2008* | | pBAD-S10Δloop |
| Recombinant DNA reagent (plasmid) | SMP252 (NEB5α+pBAD33) | *Guzman et al., 1995* | | pBAD33 |
| Recombinant DNA reagent (plasmid) | AZ321 (JM109 +pBR) | *Guillier and Gottesman, 2006* | | pBR |
| Recombinant DNA reagent (plasmid) | AZ338 (JM109 +pBR ArcZ) | *Mandin and Gottesman, 2010* | | pBR-ArcZ |
| Recombinant DNA reagent (plasmid) | AZ329 (JM109 +pBR RprA) | *Mandin and Gottesman, 2010* | | pBR-RprA |
| Recombinant DNA reagent (plasmid) | AZ417 (Top10 +pBR McaS) | *Thomason et al., 2012* | | pBR-McaS |
| Recombinant DNA reagent (plasmid) | SMP334 (MG1655 +P$_{flgB}$ GFP) | *Zaslaver et al., 2006* | | P$_{flgB}$-GFP |
| Recombinant DNA reagent (plasmid) | SMP340 (MG1655 +P$_{fliL}$ GFP) | *Zaslaver et al., 2006* | | P$_{fliL}$-GFP |
| Recombinant DNA reagent (plasmid) | SMP044 (MG1655 (crl⁻)+pBR*-UhpU) | this study | GSO1089 | pBR*-UhpU; For requests, see "Data and Materials Availability" section |
| Recombinant DNA reagent (plasmid) | SMP021 (TOP10 +pZE MotR) | this study | GSO1090 | pZE-MotR; For requests, see "Data and Materials Availability" section |
| Recombinant DNA reagent (plasmid) | SMP076 (MG1655 (crl-)+pZE-MotR*) | this study | GSO1091 | pZE-MotR*; For requests, see "Data and Materials Availability" section |
| Recombinant DNA reagent (plasmid) | SMP272 (NEB5α+pZE-MotR-M1) | this study | GSO1092 | pZE-MotR-M1; For requests, see "Data and Materials Availability" section |
| Recombinant DNA reagent (plasmid) | SMP273 (NEB5α+pZE-MotR*-M1) | this study | GSO1093 | pZE-MotR*-M1; For requests, see "Data and Materials Availability" section |
| Recombinant DNA reagent (plasmid) | SMP025 (TOP10 +pZE FliX) | this study | GSO1094 | pZE-FliX; For requests, see "Data and Materials Availability" section |

*Continued on next page*

*Continued*

| Reagent type (species) or resource | Designation | Source or reference | Identifiers | Additional information |
|---|---|---|---|---|
| Recombinant DNA reagent (plasmid) | SMP194 (MG1655 (crl⁻)+pZE-FliX-S) | this study | GSO1095 | pZE-FliX-S; For requests, see "Data and Materials Availability" section |
| Recombinant DNA reagent (plasmid) | SMP026 (TOP10 +pZE FlgO) | this study | GSO1096 | pZE-FlgO; For requests, see "Data and Materials Availability" section |
| Recombinant DNA reagent (plasmid) | SMP017 (Top10 +pXG10-SF-*rpsJ*-73aa) | this study | GSO1101 | pXG10-SF-*rpsJ-73aa*; For requests, see "Data and Materials Availability" section |
| Recombinant DNA reagent (plasmid) | SMP178 (NEB5α+pXG30-SF-*rplC*) | this study | GSO1102 | pXG30-SF-*rplC*; For requests, see "Data and Materials Availability" section |
| Recombinant DNA reagent (plasmid) | SMP167 (NEB5α+pXG10-SF-*rpsS-rplV*) | this study | GSO1103 | pXG10-SF-*rpsS-rplV*; For requests, see "Data and Materials Availability" section |
| Recombinant DNA reagent (plasmid) | SMP124 (Top10 +pXG30-SF-*rpsQ*) | this study | GSO1104 | pXG30-SF-*rpsQ*; For requests, see "Data and Materials Availability" section |
| Recombinant DNA reagent (plasmid) | SMP137 (NEB5α+pXG10-SF-*rpsJ-7aa*) | this study | GSO1105 | pXG10-SF-*rpsJ-7aa*; For requests, see "Data and Materials Availability" section |
| Recombinant DNA reagent (plasmid) | SMP152 (NEB5α+pXG10-SF-*rpsJΔleader*) | this study | GSO1106 | pXG10-SF-*rpsJΔleader*; For requests, see "Data and Materials Availability" section |
| Recombinant DNA reagent (plasmid) | SMP313 (NEB5α+pXG10-SF-*rpsJΔstemD*) | this study | GSO1107 | pXG10-SF-*rpsJΔstemD*; For requests, see "Data and Materials Availability" section |
| Recombinant DNA reagent (plasmid) | SMP317 (NEB5α+pXG10-SF-*rpsJΔstemE*) | this study | GSO1108 | pXG10-SF-*rpsJΔstemE*; For requests, see "Data and Materials Availability" section |
| Recombinant DNA reagent (plasmid) | SMP293 (NEB5α+pBAD33-3XFLAG-rpsJ) | this study | GSO1109 | pBAD33-3XFLAG-rpsJ; For requests, see "Data and Materials Availability" section |
| Software, algorithm | ImageJ software | ImageJ | http://rsb.info.nih.gov/ij | |
| Software, algorithm | EcoCyc version 20.0 | *Keseler et al., 2013* | | |
| Software, algorithm | R RCircos Package | *Zhang et al., 2013a* | https://cloud.r-project.org/web/packages/RCircos/index.html | |
| Software, algorithm | Kutools | ExtendOffice | https://www.extendoffice.com/product/kutools-for-excel.html | |
| Software, algorithm | CFX maestro analysis | Bio-Rad | Cat#12013758 | |

## Bacterial strains and growth conditions

*E. coli* MG1655 (GSO982 or GSO983) or MC4100 (GSO614) strains served as the WT strains in this study. All other bacterial strains studied here are listed in the *Key Resources Table* along with plasmids and oligonucleotides used. *E. coli* K-12 MG1655 genomic DNA was used as template to amplify mRNAs and sRNAs to be cloned into the respective constructs. Unless indicated otherwise, all strains were grown with shaking at 250 rpm at 37 °C in LB rich medium. Ampicillin (100 μg/ml), chloramphenicol (25 μg/ml), kanamycin (30 μg/ml), arabinose (0.2%), and IPTG (1 mM) were added where appropriate. Unless indicated otherwise, overnight cultures were diluted to an OD$_{600}$=0.05 and grown for the indicated times or to the desired optical densities.

Strain construction *fliA::kan*, *fliCX::kan*, and *fliX-M1:kan* strains were constructed by amplifying the *kan*$^R$ sequence from pKD4 (*Datsenko and Wanner, 2000*) using oligonucleotides listed in *Supplementary file 3* and recombining (*Datsenko and Wanner, 2000*) the product into the chromosome of

strain NM400 (kind gift of Nadim Majdalani). *flgJ* was SPA-tagged by amplifying the SPA sequence adjacent to *kan^R* sequence from pJL148 (*Zeghouf et al., 2004*) using oligonucleotides listed in *Supplementary file 3* and recombining (*Datsenko and Wanner, 2000*) the product into the chromosome of strain NM400. *motR-M1* strain was constructed using the scar-free system, FRUIT (*Stringer et al., 2012*) as previously described. Briefly, *thyA* was deleted from MG1655 (*crl*) (GSO983) strain by PCR amplification of Δ*thyA* from AMD061 *Stringer et al., 2012* followed by recombination using pKD46 (*Datsenko and Wanner, 2000*). Next, *thyA* was inserted back to the genome next to the site of mutation and selection was made by growth on minimal media lacking thymine. The *motR-M1* mutation was introduced while simultaneously removing *thyA*. The selection for colonies missing *thyA* was carried out using minimal medium M9 plates supplied with 0.4% glucose, 0.2% casamino acids, 20 µg/ml trimethoprim, and 100 µg/ml thymine. *lrhA::kan*, and *nusB::kan* deletion strains were obtained from other groups (*Baba et al., 2006*) as referenced in *Key Resources Table*. All deletions and mutations were confirmed by sequencing and then transferred to new backgrounds by P1 transduction. Where indicted, *kan^R* was removed from the chromosome using plasmid pCP20 (*Cherepanov and Wackernagel, 1995*).

Construction of strains carrying chromosomal *lacZ* fusions was carried out using PM1205 as previously described (*Mandin and Gottesman, 2009*). In brief, the *lrhA* fragment was amplified using KAPA Hifi (Fisher Scientific) using oligonucleotides SM079 and SM080 (*Supplementary file 3*) and transformed into PM1205 with a series of selective screens on minimal media plates supplemented with sucrose, LB, LB supplemented with chloramphenicol, and LB supplemented with tetracycline. Mutagenesis of *lrhA-lacZ* fusion was achieved by recombineering an *lrhA-M1* sequence instead of the WT *lrhA* sequence, using gBlock listed in *Supplementary file 3*.

## Plasmid construction

Descriptions of plasmids used in this study are in *Supplementary file 3*. Construction of the constitutive overexpression plasmids was done according to *Urban and Vogel, 2009* using pZE12-luc. The IPTG-inducible UhpU overexpression plasmid was constructed using a pBRplac derivative harboring *kan^R*, pMSG14 (*Guo et al., 2014*). The *uhpU* sequence, starting from its second nt, was amplified by PCR using oligonucleotides TU558 and TU561 (*Supplementary file 3*), digested with *Aat*II and *Hin*dIII and cloned into pMSG14 digested with the same restriction enzymes. 3XFLAG-*rpsJ* was expressed from pBAD33 (*Guzman et al., 1995*). The S10 leader and *rpsJ* sequence along with the 3XFLAG sequence was PCR amplified using oligonucleotides SM533 and SM435, digested with *Kpn*I and *Hin*dIII and cloned into pBAD33 digested with the same restriction enzymes. Construction of GFP-fusion plasmids was carried out principally as described in *Urban and Vogel, 2009*, using the pXG10-SF or pXG30-SF (*Corcoran et al., 2012*). Briefly, regions of target genes, mainly regions captured in the chimeric fragments, were PCR amplified, digested with *Mph*1103I and *Nhe*I and cloned into pXG10-SF or pXG30-SF digested with the same restriction enzymes. The full list of oligonucleotides used in this study can be found in *Supplementary file 3*. Mutagenesis of the different plasmids was achieved using the QuikChange Lightning Site-Directed Mutagenesis Kit (Agilent). All plasmids were freshly transformed into the appropriate strains before each of the experiments.

## RNA isolation

Cells corresponding to the equivalent of 10–20 OD$_{600}$ were collected, washed once with 1 X PBS, and frozen in liquid nitrogen. RNA was extracted according to the standard TRIzol protocol (Thermo Fisher Scientific) as described previously (*Melamed et al., 2020*). At the last step, RNA was resuspended in 20–50 µl of DEPC water and quantified using a NanoDrop (Thermo Fisher Scientific).

## RNA coimmunoprecipitation (Co-IP) assay

RNAs that co-IP using polyclonal antibodies to Hfq were isolated as described (*Zhang et al., 2002*) with the following modifications. MG1655 (GSO983) was grown to OD$_{600}$ ~0.6 and ~1.0 in LB medium. Cells corresponding to the equivalent of 20 OD$_{600}$ were collected, and cell lysates were prepared by vortexing with 212–300 µm glass beads (Sigma-Aldrich) in a final volume of 1 ml of lysis buffer (20 mM Tris-HCl/pH 8.0, 150 mM KCl, 1 mM MgCl$_2$, 1 mM DTT). Co-IPs were carried out using 100 µl of α-Hfq, 120 mg of protein A-Sepharose beads (GE Healthcare), and 950 µl of cell lysate. Co-IP RNA was isolated from protein A-Sepharose beads by extraction with phenol: chloroform:isoamyl alcohol

(25:24:1), followed by ethanol precipitation. Total RNA was isolated from 50 ml of cell lysate by TRIzol (Thermo Fisher Scientific) extraction followed by chloroform extraction and isopropanol precipitation. Total and co-IP RNA samples were resuspended in 15 µl of DEPC water, and 5 µg total RNA and 0.5 µg co-IP RNA were subjected to northern analysis as described below.

## Northern blot analysis

For smaller RNAs, total RNA (5 µg) was separated on a denaturing 8% polyacrylamide urea gel containing 6 M urea (1:4 mix of Ureagel Complete to Ureagel-8 (National Diagnostics) with 0.08% ammonium persulfate) in 1 X TBE buffer at 300 V for 90 min. The RNA was transferred to a Zeta-Probe GT membrane (Bio-Rad) at 20 V for 16 hr in 0.5 X TBE. For longer RNAs, total RNA (10 µg) was fractionated on formaldehyde-MOPS agarose gels as previously described (*Adams et al., 2017*). Briefly, RNA was denatured in 3.7% formaldehyde (Fisher), 1 X MOPS (20 mM MOPS, 5 mM NaOAc, 1 mM EDTA, pH 7.0) and 1 X RNA loading dye (Thermo Fisher Scientific) for 10 min at 70 °C and incubated on ice. The RNA was loaded onto a 2% NuSieve 3:1 agarose (Lonza), 1 X MOPS, 2% formaldehyde gel and separated at 125–150 V at 4 °C for 1–2 hr and then transferred to a Zeta-Probe GT membrane (Bio-Rad) via capillary action overnight (*Streit et al., 2009*). For both types of blots, the RNA was crosslinked to the membranes by UV irradiation. RiboRuler High Range and Low Range RNA ladders (Thermo Fisher Scientific) were marked by UV-shadowing. Oligonucleotide probes (listed in *Supplementary file 3*) for the different RNAs were labelled with 0.3 mCi of [γ-$^{32}$P] ATP (Perkin Elmer) by incubating with 10 U of T4 polynucleotide kinase (New England Biolabs) at 37 °C for 1 hr.

## Primer extension assay

Primer extension analysis was performed using an oligonucleotide (listed in *Supplementary file 3*) specific to the *rpsS* as described (*Zhang et al., 1998*). RNA samples (5 µg of total RNA) were incubated with 2 pmol of 5-$^{32}$P-end-labeled oligonucleotide primer at 80 °C and then slow-cooled to 42 °C. After the addition of dNTPs (1 mM each) and AMV reverse transcriptase (10 U, Life Sciences Advanced Technologies Inc), the reactions were incubated in a 10 µl-reaction volume at 42 °C for 1 hr. The reactions were terminated by adding 10 µl of Stop Loading Buffer. The cDNA products then were fractionated on 8% polyacrylamide urea gels containing 6 M urea in 1 X TBE buffer at 70 W for 70 min.

## RT-qPCR

Total RNA was isolated from cultures grown to OD$_{600}$~0.2 and RNA concentrations were determined using a NanoDrop (Thermo Fisher Scientific). Samples were treated with DNase using TURBO DNA-free Kit (Thermo Fisher Scientific). DNA-free RNA was used for cDNA synthesis using iScript cDNA Synthesis Kit (Bio-Rad) and cDNA concentrations were measured by Qubit fluorimeter (Invitrogen). Equal amounts of cDNA were loaded into 96-well plate and cDNA was quantified by CFX Connect Real-Time system (Bio-Rad) using iTaq Univer SYBR Green mix (Bio-Rad) according to manufacturer instructions. Specific oligonucleotide primers were designed for each gene and the expression was normalized using *ssrA* levels. Serial dilutions of *E. coli* genomic DNA in known concentrations were used to generate a standard curve. CFX maestro analysis software (Bio-Rad) was used to determine the starting quantities of the cDNA samples based on the standard curve, and normalization was done using the starting quantities of *ssrA*. Reactions for each biological replicate were performed in technical duplicate or triplicate.

## RNA structure probing

gBlock fragments carrying the *motR*, *fliX*, *rpsJ* or *rpsS* CDS (IDT) were used as DNA templates for in vitro transcription with MEGAshortscript T7 High Yield Transcription Kit (Invitrogen). The transcripts were dephosphorylated with calf intestinal alkaline phosphatase (CIP, New England Biolabs) and then radioactively labeled at 5′ end with [γ-$^{32}$P] ATP (Perkin Elmer) and T4 kinase (Invitrogen), and purified on an 8% polyacrylamide/6 M urea gel and eluted in buffer containing 20 mM Tris-HCl/pH 7.5, 0.5 M NaOAc, 10 mM EDTA and 1% SDS at 4 °C for overnight, followed by ethanol precipitation. The RNA concentration was determined by measuring the OD$_{260}$ on Nanodrop (Thermo Fisher Scientific).

For all the structural probing assays, 0.2 pmole of the labeled transcript, 2 pmole of unlabeled transcript and 1 µg of yeast RNA with or without 2 pmole (hexameric concentration) of purified Hfq were mixed in 10 µl of 1 x Structural Buffer in Ambion RNase T1 Kit (Invitrogen). The reactions were

incubated at 37 °C for 10 min, followed by treatment at 37 °C with 0.02 U RNase T1 for 10 min, 1.3 U RNase III for 1.5 min, or 50 µmole lead acetate for 10 min, whereupon 20 µl Inactivation Buffer and 1 µl Glycoblue were added. The RNAs were precipitated and resuspended in 10 µl Gel Loading Buffer II (Thermo Fisher Scientific), and analyzed on a 8% polyacrylamide/7 M urea gel run in 1 x TBE. RNase T1 and alkali digestion ladders of the end-labeled transcripts were used as molecular size markers.

## Translational reporter assays

The GFP reporter assays were carried out essentially as described (*Melamed et al., 2016*). Overnight cultures were grown in 2 ml of LB media supplemented with the appropriate antibiotics at 37 °C with constant shaking at 250 rpm. Cells were then diluted to $OD_{600}$~0.05 in 1 ml of fresh LB medium supplemented with the appropriate antibiotics in 96-well plate and grown at 37 °C with constant shaking at 250 rpm for 3 hr. Cells were pelleted and resuspended in filtered 1 X PBS. Fluorescence was measured using the BD LSRFortessa or Beckman Coulter Cytoflex flow cytometer. The level of regulation was calculated by subtracting the auto-fluorescence and then calculating the ratio between the fluorescence signal of a strain carrying the sRNA over-expressing plasmid and the signal of a strain carrying the control plasmid. Three biological repeats were prepared for every sample.

The β-galactosidase assays were carried out as described (*Miller, 1992*). Overnight cultures grown as for the GFP reporter assays were diluted 1:100 into 5 ml of fresh LB with antibiotic and 0.2% arabinose and grown at 37 °C with constant shaking at 250 rpm until $OD_{600}$ ~0.7. IPTG (1 mM) was added to cells harboring inducible sRNAs plasmids. After β-galactosidase activity was measured, the Miller units were calculated from the following formula:

$$Miller\ Unit = \frac{1000\left(OD_{420} - 1.75 \cdot OD_{550}\right)}{t_{min} \cdot OD_{600}}$$

## Transcriptional reporter assays

Overnight cultures harboring *flgB-gfp* and *fliL-gfp* fusions (*Zaslaver et al., 2006*) were grown as described for the translation reporter assays and then diluted to $OD_{600}$~0.05 in 150 µl of fresh LB medium supplemented with the appropriate antibiotics in a transparent bottom 96-well plate. Bacterial growth and promoter activity were monitored for 330 min at 37 °C using $OD_{600}$ and GFP fluorescent measurements, respectively, using a Synergy H1 plate reader (Agilent).

## Immunoblot analysis

Bacteria were grown to the desired $OD_{600}$, and the cells in 0.5 ml – 4 ml of culture were collected. Cell lysates were prepared by resuspending cell pellets with Laemmli sample buffer (Bio-Rad) normalized to the cell density, and samples were then heated for 10 min at 95 °C. Protein samples were subjected to a 4–15% polyacrylamide SDS gel electrophoresis followed by electrotransfer to a nitrocellulose membrane (Fisher Scientific). The membrane was blocked with 3% milk in 1X PBS with 0.1% Tween 20 (PBST), probed with anti-flagellin antibodies (1/10,000) (Abcam) and then with anti-rabbit secondary antibody (1/10,000) or with ANTI-FLAG M2-Peroxidase (HRP) (1/1000), (Sigma-Aldrich). Signals were visualized by the ECL system (Bio-Rad).

## Flagellin measurements

Δ*fliC* (GSO1073) or WT (GSO983) cells harboring pBR*, pBR*-UhpU, pZE, pZE-MotR, pZE-MotR*, pZE-FliX or pZE-FliX-S were grown with shaking at 180 rpm in 5 ml of LB at 37 °C to $OD_{600}$ ~1.0. Cell pellets collected by centrifugation were suspended in 5 ml of PBS and then heated at 65 °C for 5 min, followed by centrifugation to obtain the cell pellets and supernatants, which contained the cytoplasmic flagellin molecules and depolymerized flagellin monomers, respectively. The cell pellets were resuspended in the Laemmli sample buffer (Bio-Rad), normalized to the cell density. Proteins in the supernatants were precipitated by 10% trichloroacetic acid, resuspended in Laemmli sample buffer (Bio-Rad) and heated at 95 °C for 10 min.

## Electron microscopy

Overnight cultures were diluted in fresh medium and grown with shaking at 180 rpm, at 37 °C to mid-log phase ($OD_{600}$~0.6–0.8) unless indicated otherwise. Cells were collected by centrifugation at

1000 rpm for 20 min, and pellet was resuspended in 300 µl of saline. Next, 3 µl of bacterial suspension were placed on a freshly glow-discharged carbon covered electron microscopic support grid (EMS, Hatfield, PA) for 5 min. The grid was washed twice with distilled water and stained for 1 min with 0.75% aqueous solution of uranyl formate, pH 4.5. The grids were imaged in Thermo Fisher Scientific (Hillsboro, OR) FEI Tecnai 20 electron microscope operated at 120 kV. The images were recorded using AMT (Woburn, MA) XR81 CCD camera. Flagella were counted for 20–40 cells in each sample as indicated in the Figure legends. Each analysis was repeated a minimum of three times.

## Motility assays

Overnight cultures (~1 µl) were spotted onto 0.3% soft agar plates by touching the agar softly with the tip and ejecting the culture. Plates were incubated right-side up at 30 °C above a beaker filled with water for 9–24 hr. Plates were made with the appropriate antibiotics and with 1 mM IPTG when needed. The plates were imaged using Bio-Rad imager (using Colorimetric settings) and the diameter of the bacterial culture was calculated using ImageJ software. Two technical repeats and three biological repeats were carried out for each strain. For motility competition assays, cells were first transformed with pCON1.proC-GFP or pCON1.proC-mCherry plasmids (*Cooper et al., 2017*), resulting in a GFP or an mCherry signal, respectively. In each case, equal numbers of bacterial cells based on $OD_{600}$ of each overnight culture for one strain expressing a green fluorescence signal and a second strain expressing a red fluorescent signal were mixed before spotting them onto 0.3% soft agar plate and the plates were incubated as described above. Images were taken using Bio-Rad imager with the following settings: Colorimetric (1–2 s) for bright field, Cy2 for GFP (auto optimal exposure), Cy3 for mCherry (auto optimal exposure). Images were merged using Image Lab (Bio-Rad).

## RNA-seq

Overnight cultures were diluted in fresh LB medium and grown to early-log phase ($OD_{600}$~0.2). RNA was extracted using the standard TRIzol protocol (Thermo Fisher Scientific) as described above. Total RNA libraries were constructed using the RNAtag-Seq protocol with a few modifications to allow capture of short RNA fragments as previously described (*Melamed et al., 2018*). The libraries were sequenced by paired-end sequencing using the HiSeq 2500 system (Illumina) at the Molecular Genomics Core, *Eunice Kennedy Shriver* National Institute of Child Health and Human Development. RNA-seq data processing followed the same procedures as RIL-seq data analysis for QC analysis, adaptor removal, and alignment with the Python RILSeq package (*Melamed et al., 2018*). The raw fastq records were demultiplexed with python script index_splitter.py (https://github.com/asafpr/RNAseq_scripts/blob/master/index_splitter.py; *Peer, 2015*) followed by adapter removal with cutadpt software (version 3.4). The trimmed fastq reads were mapped to the *E. coli* genome (ecoli-k12-MG1655-NC_000913–3) with Python RILSeq package (version 0.74, https://github.com/asafpr/RILseq; *Peer, 2018*). Deeptools software (version 3.5.1) was used to generate bigwig file for coverage visualization. Read counts were obtained with featureCounts tool of Subread software (version 2.0.3) and a customized annotation file based on EcoCyc version 20.0 (*Keseler et al., 2013*) with manual addition of sRNAs and small proteins from *Hör et al., 2020*; *Hemm et al., 2020*. Differential expression analyses were conducted with R DESeq2 package (*Love et al., 2014*) and default normalization. Differentially-expressed genes were extracted with the parameter of 'independent-Filtering = FALSE'.

## Determination of sequence motifs and base-pairing predictions

Common binding motifs were searched with MEME software (*Bailey et al., 2009*). Genes that were induced the most by MotR* overexpression in RNA-seq data (*Supplementary file 2*) (FDR = 0.05 and ≥2 fold) were extracted and grouped into transcription units based on EcoCyc version 20.0 (*Keseler et al., 2013*). For each transcription unit, genomic sequence was extracted using coordinates for the start codon of the first gene in the transcription unit and 250 nt upstream of the gene. For sRNAs, genomic sequence was extracted using coordinates for the transcription start site and 250 nt upstream to the gene. For outputs, motif length was restricted to 28 nt. Base-pairing regions between two RNAs were predicted using IntaRNA (*Mann et al., 2017*) or TargetRNA2 (*Kery et al., 2014*).

## Functional annotation analysis

Functional annotation analysis of sRNAs targets was carried out using the Database for Annotation, Visualization and Integrated Discovery (DAVID) (*Huang et al., 2009*). Gene names served as the input list in each case. Targets that were present in at least three RIL-seq conditions in *Supplementary file 1* were included in the analysis.

## Circos plots

Circos plots were generated according to the R RCircos Package (*Zhang et al., 2013a*). Link lines are used to label the statistically significant chimeric fragments (S-chimeras as defined in *Melamed et al., 2016*). RIL-seq data from six different growth conditions was analyzed and S-chimeras present in at least four of the six conditions are included in the plots.

## Browser images

Data from RIL-seq experiment 1 from *Melamed et al., 2020* extracted from unified S-chimera files for the different sRNAs were mapped based on the first nt of each read in the chimera. BED files were generated with Python RILSeq package (*Melamed et al., 2018*) and viewed using the UCSC genome browser (*Kent et al., 2002*). For previously annotated RNA in GTF file, BED files are directly generated with command of generate_BED_file_of_endpoints.py and EcoCyc ID. For genes annotated in the current study, significant chimeras which involve the relevant gene are first extracted from significant interaction file, then chimeric reads involving the S-chimeras are extracted from chimeric read file. To be a qualified chimeric read, RNA1 start position of the read must overlap with the genomic range of RNA1 in S-chimera and RNA2 start position of the read must overlap with the genomic range of RNA2 in S-chimera. Finally, the read list for genes annotated in the current study is supplied to generate_BED_file_of_endpoints.py command to generate BED file.

## Data and materials Availability

Further information and requests for resources and reagents should be directed to and will be fulfilled by the Lead Contact, Gisela Storz (storzg@mail.nih.gov). The sequencing data reported in this paper have been deposited in GEO under accession number GSE174487. Reused sequencing data from *Melamed et al., 2016*; *Melamed et al., 2020* have been deposited in ArrayExpress under accession number E-MTAB-3910 and in GEO under accession number GSE131520.

## Acknowledgements

We thank M Gottesman for plasmids expressing wild type and *rpsJ* mutants, O Steele-Mortimer for plasmids constitutively expressing GFP or mCherry, and D Court for the S10 antibody. We thank J Wade for sharing the sequences used to generate the σ[28] binding motif and J Wade and G Baniulyte for advice on the FRUIT method. We thank the NICHD Molecular Genomics Core, particularly Tianwei Li, for all the library sequencing. We also appreciate the help of A Peer with the sRNA conservation analysis. We are grateful to the Storz and S Gottesman labs for all the helpful discussions and thank the Storz lab, S Gottesman, and J Wade for their comments on the manuscript. This work utilized the computational resources of the NIH HPC Beowulf cluster (http://hpc.nih.gov).

# Additional information

### Funding

| Funder | Grant reference number | Author |
| --- | --- | --- |
| Israel Science Foundation | 826/22 | Sahar Melamed |
| Israel Science Foundation | 2859/22 | Sahar Melamed |
| National Institutes of Health | 1ZIAHD001608-32 | Gisela Storz |

| Funder | Grant reference number | Author |
|---|---|---|

The funders had no role in study design, data collection and interpretation, or the decision to submit the work for publication.

## Author contributions

Sahar Melamed, Conceptualization, Data curation, Formal analysis, Supervision, Funding acquisition, Validation, Investigation, Methodology, Writing – original draft, Project administration, Writing – review and editing; Aixia Zhang, Data curation, Formal analysis, Investigation, Visualization, Methodology, Writing – original draft, Writing – review and editing; Michal Jarnik, Data curation, Formal analysis, Investigation, M.J. performed all EM analysis; Joshua Mills, Aviezer Silverman, Data curation, Formal analysis, Investigation; Hongen Zhang, Resources, Data curation, Software, Formal analysis, H.Z. performed all computational analyses; Gisela Storz, Conceptualization, Supervision, Funding acquisition, Investigation, Visualization, Writing – original draft, Project administration, Writing – review and editing

## Author ORCIDs

Sahar Melamed http://orcid.org/0000-0003-1360-1297
Aixia Zhang https://orcid.org/0000-0003-0540-5990
Michal Jarnik https://orcid.org/0000-0002-1269-3508
Joshua Mills http://orcid.org/0000-0003-3179-5566
Hongen Zhang https://orcid.org/0000-0001-6871-8463
Gisela Storz http://orcid.org/0000-0001-6698-1241

Reviewer #1 (Public Review): https://doi.org/10.7554/eLife.87151.3.sa1
Reviewer #2 (Public Review): https://doi.org/10.7554/eLife.87151.3.sa2
Reviewer #3 (Public Review): https://doi.org/10.7554/eLife.87151.3.sa3
Author Response https://doi.org/10.7554/eLife.87151.3.sa4

# Additional files

## Supplementary files

• Supplementary file 1. Target sets of σ$^{28}$-dependent sRNAs based on RIL-seq datasets. RIL-seq datasets from experiments done in six different conditions (*Melamed et al., 2020*; *Melamed et al., 2016*) were analyzed tin order to generate a target set for each of the four sRNAs. Tables are sorted according to the number of conditions in which a target was found. Functional annotation analysis of sRNA targets was done using DAVID. Targets that were present in at least three RIL-seq conditions in were included in the analysis. The top annotation cluster is shown for each dataset. (NOR = Normalized Odds Ratio)

• Supplementary file 2. RNA levels in different RNAseq datasets. Total RNA libraries reads were subject to differential expression analyses conducted with DESeq2 (*Love et al., 2014*). For MotR* and FliX data, three biological repeats were analyzed for the vector control strain (pZE) and for the MotR* or FliX overexpressing strain (pZE-MotR*, pZE-FliX). For Δ*fliA* data (*Fitzgerald et al., 2014*), two biological repeats were analyzed for the WT strain and for the Δ*fliA* strain.

• Supplementary file 3. List of oligonucleotides used in this work.

• MDAR checklist

## Data availability

The sequencing data reported in this paper have been deposited in GEO under accession number GSE174487.

The following dataset was generated:

| Author(s) | Year | Dataset title | Dataset URL | Database and Identifier |
|---|---|---|---|---|
| Melamed S, Zhang A, Jarnik M, Mills J, Silverman A, Zhang H, Storz G | 2023 | σ28-dependent small RNA regulation of flagella biosynthesis | https://www.ncbi.nlm.nih.gov/geo/query/acc.cgi?acc=GSE174487 | NCBI Gene Expression Omnibus, GSE174487 |

The following previously published datasets were used:

| Author(s) | Year | Dataset title | Dataset URL | Database and Identifier |
|---|---|---|---|---|
| Melamed S, Peer A, Faigenbaum-Romm R, Gatt YE, Reiss N, Bar A, Altuvia Y, Argaman L, Margalit H | 2016 | Global mapping of small RNA-target interactions in bacteria | https://www.ebi.ac.uk/biostudies/arrayexpress/studies/E-MTAB-3910?accession=E-MTAB-3910 | ArrayExpress, E-MTAB-3910 |
| Melamed S, Adams PP, Zhang A, Zhang H, Storz G | 2020 | RNA-RNA interactomes of ProQ and Hfq reveal overlapping and competing roles | https://www.ncbi.nlm.nih.gov/geo/query/acc.cgi?acc=GSE131520 | NCBI Gene Expression Omnibus, GSE131520 |

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
