## [Editor Report · eLife assessment]

This article provides **important** findings on how bacteria use small RNAs to regulate flagellar expression with implications for multiple fields. The data supporting the conclusions are **convincing** with a large amount of data that include results from phenotypic analyses, genomics approaches as well as in-vitro and in-vivo target identification and validation methods. This study on the varied effects of three sRNAs (UhpU, FliX and MotR) is of broad interest to RNA biochemists and microbiologists.

---

## [Referee Report · Reviewer #1 (Public Review)]

Bacteria can adapt to extremely diverse environments via extensive gene reprogramming at transcriptional and post-transcriptional levels. Small RNAs are key regulators of gene expression that participate in this adaptive response in bacteria, and often act as post-transcriptional regulators via pairing to multiple mRNA-targets.

In this study, Melamed et al. identify four *E. coli* small RNAs whose expression is dependent on sigma 28 (FliA), involved in the regulation of flagellar gene expression. Even though they are all under the control of FliA, expression of these 4 sRNAs peaks under slightly different growth conditions and each has different effects on flagella synthesis/number and motility. Combining RILseq data, structural probing, northern-blots and reporter assays, the authors show that 3 of these sRNAs control fliC expression (negatively for FliX, positively for MotR and UhpU) and two of them regulate r-protein genes from the S10 operon (again positively for MotR, and negatively for FliX). UhpU also directly represses synthesis of the LrhA transcriptional regulator, that in turn regulates flhDC (at the top of flagella regulation cascade). Based on RILseq data, the fourth sRNA (FlgO) has very few targets and may act via a mechanism other than base-pairing.

As r-protein S10 is also implicated in anti-termination via the NusB-S10 complex, the authors further hypothesize that the up-regulation of S10 gene expression by MotR promotes expression of the long flagellar operons through anti-termination. Consistent with this possible connection between ribosome and flagella synthesis, they show that MotR overexpression leads to an increase in flagella number and in the mRNA levels of two long flagellar operons, and that both effects are dependent on the NusB protein. Lastly, they provide data supporting a more general activating and repressing role for MotR and FliX, respectively, in flagellar genes expression and motility, via a still unclear detailed mechanism.

This study brings a lot of new information on the regulation of flagellar genes, from the identification of novel sigma 28-dependent sRNAs to their effects on flagella production and motility. It represents a considerable amount of work; the experimental data are clear and solid and support the conclusions of the paper. Even though mechanistic details underlying the observed regulations by MotR or FliX sRNAs are lacking, the effect of these sRNAs on fliC, several rps/rpl genes, and flagellar genes and motility is convincing.

The connection between r-protein genes regulation and flagellar operons is exciting, and so is the general effect of pMotR or pFliX on the expression of multiple middle and late flagellar genes.

---

## [Referee Report · Reviewer #2 (Public Review)]

This manuscript discusses the posttranscriptional regulation of flagella synthesis in *Escherichia coli*. The bacterial flagellum is a complex structure that consists of three major domains, and its synthesis is an energy-intensive process that requires extensive use of ribosomes. The flagellar regulon encompasses more than 50 genes, and the genes are activated in a sequential manner to ensure that flagellar components are made in the order in which they are needed. Transcription of the genes is regulated by various factors in response to environmental signals. However, little is known about the posttranscriptional regulation of flagella synthesis. The manuscript describes four UTR-derived sRNAs (UhpU, MotR, FliX, and FlgO) that are controlled by the flagella sigma factor σ28 (fliA) in *Escherichia coli*. The sRNAs have varied effects on flagellin protein levels, flagella number, and cell motility, and they regulate different aspects of flagella synthesis.

UhpU corresponds to the 3´ UTR of uhpT.

UhpU is transcribed from its own promoter inside the coding sequence of uhpT.

MotR originates from the 5´ UTR of motA. The promoter for motR is within the flhC CDS and is also the promoter of the downstream motAB-cheAW operon.

FliX originates from the 3´ UTR of fliC. Probably processed from parental mRNA.

FlgO originates from the 3´ UTR of flgL. Probably processed from parental mRNA.

This is a very interesting study that shows how sRNA-mediated regulation can create a complex network regulating flagella synthesis. The information is new and gives a fresh outlook at cellular mechanisms of flagellar synthesis.

---

## [Referee Report · Reviewer #3 (Public Review)]

Flagella are crucial for bacterial motility and virulence of pathogens. They represent large molecular machines that require strict hierarchical expression control of their components. So far, mainly transcriptional control mechanisms have been described to control flagella biogenesis. While several sRNAs have been reported that are environmentally controlled and regulate motility mainly via control of flagella master regulators, less is known about sRNAs that are co-regulated with flagella genes and control later steps of flagella biogenesis.

In this carefully designed and well-written study, the authors explore the role of four *E. coli* σ28-dependent 3' or 5' UTR-derived sRNAs in regulating flagella biogenesis. UhpU and MotR sRNAs are generated from their own σ28(FliA)-dependent promoter, while FliX and FlgO sRNAs are processed from the 3'UTRs of flagella genes under control of FliA. The authors provide an impressive amount of data and different experiments, including phenotypic analyses, genomics approaches as well as in-vitro and in-vivo target identification and validation methods, to demonstrate varied effects of three of these sRNAs (UhpU, FliX and MotR) on flagella biogenesis and motility. For example, they show different and for some sRNAs opposing phenotypes upon overexpression: While UhpU sRNA slightly increases flagella number and motility, FliX has the opposite effect. MotR sRNA also increases the number of flagella, with minor effects on motility.

While the mechanisms and functions of the fourth sRNA, FlgO, remain elusive, the authors provide convincing experiments demonstrating that the three sRNAs directly act on different targets (identified through the analysis of previous RIL-seq datasets), with a variety of mechanisms. The authors demonstrate, UhpU sRNA, which derives from the 3´UTR of a metabolic gene, downregulates LrhA, a transcriptional repressor of the flhDC operon encoding the early genes that activate the flagellar cascade. According to their RIL-seq data analyses, UhpU has hundreds of additional potential targets, including multiple genes involved in carbon metabolism. Due to the focus on flagellar biogenesis, these are not further investigated in this study and the authors further characterize the two other flagella-associated sRNAs, FliX and MotR. Interestingly, they found that these sRNAs seem to target coding sequences rather than acting via canonical targeting of ribosome binding sites. The authors show FliX sRNA represses flagellin expression by interacting with the CDS of the fliC mRNA. Both FliX and MotR sRNA turn out to modulate the levels of ribosomal proteins of the S10 operon with opposite effects. MotR, which is expressed earlier, interacts with the leader and the CDS of rpsJ mRNA, leading to increased S10 protein levels and S10-NusB complex mediated anti-termination, promoting readthrough of long flagellar operons. FliX interacts with the CDSs of rplC, rpsQ, rpsS-rplV, repressing the production of the encoded ribosomal proteins. The authors also uncover MotR and FliX affect transcription selected representative flagellar genes, with an unknown mechanism.

Overall, this comprehensive study expands the repertoire of characterized UTR derived sRNAs and integrates new layers of post-transcriptional regulation into the highly complex flagellar regulatory cascade. Moreover, these new flagella regulators (MotR, FliX) act non-canonically, and impact protein expression of their target genes by base-pairing with the CDS of the transcripts. Their findings directly connect flagella biosynthesis and motility, highly energy consuming processes, to ribosome production (MotR and FliX) and possibly to carbon metabolism (UhpU). In their revised version, the authors have addressed many of the previously raised questions and comments. This made their manuscript easier to read and to follow.

---

## [Author Response]

The following is the authors’ response to the original reviews.

**Reviewer #1 (Recommendations For The Authors):**
p. 5, l. 87-90: The control of flgM by OmrA/B (PMID 32133913) and the antisense RNA to flhD (PMID 36000733) are other examples of known regulatory RNAs that impact the flagellar regulon.

We thank the reviewer for pointing out these references and have added citations to them (page 5, lines 87-91).

p.11/Fig. 3: it is intriguing that ArcZ and RprA, two of the rpoS-activating sRNAs, repress lrhA. I realize that it is outside of the scope of this study, but have the authors considered the possibility that ArcZ or McaS could have a role in the previously reported repression of rpoS by LrhA (PMID 16621809)?

We agree that it is intriguing that ArcZ and RprA, two of the rpoS-activating sRNAs, repress lrhA, and added mention of this regulatory connection (page 12, lines 247-250).

p. 13/l. 272: I do not understand why the authors say that "r-proteins were almost exclusively found in chimeras with MotR and FliX and no other sRNAs...", given that several other chimeras between r-prot and other sRNAs are found

While some r-proteins encoding genes were found with other sRNAs in RIL-seq datasets, MotR and FliX generally had the highest numbers. The text was revised to better describe the RIL-seq data for r-proteins interaction partners (page 14, lines 291-295), and a new panel showing the S10 operon with all the interacting sRNAs was added to Figure 3—figure supplement 1B.

Fig. 4 and 5: One possible improvement would be to more systematically assess the effect of base-pairing mutants of the sRNAs, such as MotRM1 or FliXM1 on fliC and rps/rpl genes in vivo. This is especially important for the mutants that affected the sRNA effects in the in vitro probing assays, such as UhpU-M2, MotR-M1 and FliX-S-M1 on fliC (Fig. S7)

As suggested, we examined fliC mRNA levels across growth in motR-M1 and fliX-M1 chromosomal mutants. The results of these northern assays, now shown in Figure 8—figure supplement 1, are consistent with our model as we observed delayed expression of fliC mRNA in motR-M1 background and premature expression in fliX-M1 background (page 21, lines 444446, 449-453).

Fig. 5: it may be worth including a schematic of the whole S10 operon to highlight its length and its organization?

As suggested, a schematic representation of the S10 operon was added to Figure 3—figure supplement 1 with a summary of the RIL-seq data for this operon.

Probing data (Fig. 5, S7 and S9): in general, it is difficult to differentiate the thin and thick brackets, and what is indicated by the dashed brackets is not always clear. Maybe using a color-code instead could help? Highlighting the predicted pairing regions on the different gels could be useful as well.

We thank the reviewer for this suggestion and color-coded the brackets (Figure 5, Figure 4figure supplement 2, and Figure 5-figure supplement 2). The correspondences to regions of predicted pairing are described in the figures legends.

Fig. S10: The experimental evidence used to support FliX-dependent degradation of the rpsS mRNA is indirect (primer extension to observe higher levels of cleavage intermediates). It would be nice to be able to observe a decrease in the mRNA levels as well, either by Northern, or primer extension from a region more distant to the FliX pairing site.

The S10 operon is long (~5 KB). We have tried multiple probes for this mRNA and detect many bands with each, likely due to extensive regulation of this operon. We think teasing out the origin of the different bands to appropriately interpret changes in patterns will require a significant amount of work.

legend of Fig. S10: from the gel, it seems that only the plasmids differ in the samples, and it is not clear where the data corresponding to the WT strain mentioned in the legend is shown

The samples shown in this figure are all for the indicated plasmids in the WT strain. We corrected the figure legend.

Table S1: please define the NOR (normalized odds ratio?)

The definition of Normalized Odds Ratio was added to the legend of Supplementary file 1.

**Reviewer #2 (Recommendations For The Authors):**
Major comments:Figure 1B. Please add a negative control (which could be in the supplementary section) from a large section showing transcripts that are not directly influenced by Hfq.

We think the flgKLO browser in this figure serves as a negative control; flgK and flgL clearly are not enriched on Hfq in contrast to FlgO. Figure 1B was generated using published datasets that are easily accessible to the readers at a genome browser and show many other examples of transcripts that are not influenced by Hfq:https://genome.ucsc.edu/cgi-bin/hgTracks?hubUrl=https://hpc.nih.gov/~NICHD-core0/storz/trackhubs/ecoli_rilseq/hub.hub.txt&hgS_loadUrlName=https://hpc.nih.gov/~NICHDcore0/storz/trackhubs/ecoli_rilseq/session.txt&hgS_doLoadUrl=submit

Line 158. MotR* is a more abundant version of [the constitutively overexpressed] MotR. Is there a Northern or qPCR to confirm this? While I understand the relevance of these mutated constructs, their high expression can lead to artefactual effects.

This is a valuable point and therefore we provided a northern blot to document the relative levels of MotR and MotR* (Figure 2—figure supplement 1A).

Figure 2. The overexpression of MotR/MotR* from a plasmid is increasing the number of flagella. However, when the MotR gene is deleted, is there a reduction of the number of flagella? Same question with FliX: what happens when the fliX gene is deleted? According to the model described in the manuscript, we should expect fewer flagella in ΔmotR background and an increased number of flagella in ΔfliX background. Both Figure 2 and Figure 8 would benefit from additional experiments with deleted motR and fliX genes.

We agree that experiments regarding the endogenous effects of endogenous sRNAs are important. We provided such data in Figure 8 and Figure 8—figure supplement 1 for MotR and FliX in a variety of assays: flagella numbers by electron microscopy, motility and competition assays, expression of flagellar genes by RT-qPCR and western analysis. The chromosomallyexpressed MotR-M1 and FliX-M1 base pairing mutants did show the expected phenotypes of reduced and increased numbers of flagella, respectively (Figure 8A-B). As suggested by reviewer 1, we added northern analysis that examined fliC mRNA levels across growth in motRM1 and fliX-M1 chromosomal mutants. The results of these northern assays are consistent with our model as we observed delayed expression of fliC mRNA in motR-M1 background and premature expression in fliX-M1 background. We went to the trouble of constructing strains carrying point mutations in the chromosomal copies of these genes rather than deletions to avoid interfering with the expression of motA and fliC given that MotR and FliX encompass the 5’ and 3’ UTRs, respectively.

Figure 3 is key to demonstrating the sRNAs pairing with their specific targets and potential effect on bacterial swimming. However, these results would be more relevant with endogenous expression of the sRNAs and demonstration of their effects on the same targets. A Northern blot showing the overproduced sRNA level compared to endogenous sRNA level could help us appreciate the expression ratio.

The levels of the UhpU, MotR and FliX expressed from the overexpression plasmids are at least 100-fold higher than the endogenous levels. Thus, we agree that assays of chromosomal deletion/point mutants are important experiments. We did construct chromosomal uhpU-M1 and uhpU∆seed sequence mutants. However, under the conditions assayed, the uhpU chromosomal mutations did not result in observable effects on motility or FlhD-SPA protein levels. It is possible we would be able to detect differences between the wild type and uhpU chromosomal mutant strains under different growth conditions or in different assays, but this would require a significant amount of work. For many other sRNA chromosomal mutations have no or only subtle effects, suggesting redundancy between sRNAs or sRNA roles in fine tuning gene expression.

Figure 4. In panel B, the empty plasmid pZE alone seems to positively affect the flagellin expression when compared to the WT background. This can also be seen in Figure 4C. There is no fliC signal with empty plasmid pBR* but a strong fliC signal with empty plasmid pZE. Maybe the authors can explain this in the manuscript.

With respect to panel B and Figure 4—figure supplement 1A, we agree that there is some variation between the levels of flagellin in the WT and pZE control samples, possibly due to the addition of antibiotic to the pZE culture. We added quantification of the bands in Figure 4— figure supplement 1 to better document the changes in flagellin levels.

With respect to panel C, the pBR* samples were collected in crl+ background while the pZE samples were collected in crl- background, which explains the lack of fliC signal in the pBR* control sample. This is now noted in the figure legend.

In lines 154-157, the justification for using two plasmids is described. An IPTG-inducible Plac promoter, the pBR*, is used because the constitutive overexpression of UhpU is resulting in mutated UhpU clones. These observations suggest a toxic expression level of UhpU that the cell can only tolerate when the UhpU RNA is somewhat deactivated by mutations. This does not seem like a detail and could be discussed further.

We agree with the reviewer that this observation is important and now mention that it suggests at a critical UhpU role (page 8, lines 160-163).

Figure 5E and I. While the bindings of MotR on rpsJ and Flix-S on rpsS are clear, the resolution of both gels in the areas of binding (upper part of both gels) could be improved.

We found it tricky to choose the mRNA fragments for the in vitro structure probing for the regions of predicted pairing internal to CDSs. Given that we hoped to retain native RNA folding, we chose long fragments; for rpsJ, we started with the +1 of S10 leader and for rpsS, we started 147 nt into the CDS, a region that overlaps the region that was cloned to the rpsS-rplV-gfp fusion. Consequently, the region of base pairing is in the upper part of both gels. The gels were already run for an unusually long time. Thus, we do not think the resolution could be improved further. Nevertheless, we think the region of protection is evident for both mRNAs.

Minor comments:Fig 1B. The promoter symbols are extremely small, please increase the size.

As suggested, we have enlarged the promoter symbols in Figure 1B as well as in Figure 3A.

Line 211. "the lrhA mRNA has an unusually long 5´ UTR". How long exactly?

The 5’ UTR of the lrhA mRNA is 371 nt long. This is now mentioned in the text (page 11, line 224)

Line 320. Should "Fig 9C" be "Fig S9C" instead?

We thank the reviewer for noticing this typo. Callouts to supplementary figures have now been renumbered per eLife format.

Line 384. Something seems to be missing in the sentence "a representative combined class 2 and 3 promoter".

The sentence has been modified to clarify the designation (page 19, lines 409-411).

**Reviewer #3 (Recommendations For The Authors):**
Recommendation to clarify/strengthen the presentation of science in the paper:Lines 102-103: Can the authors provide some more information on how the sRNAs were initially discovered to be potentially sigma-28 dependent and selected?

As suggested, we expanded the section discussing the discovery and the selection of these sRNAs (page 6, lines 104-109).

Lines 192-193: It would be helpful to provide a bit more information in the main text about what are the different RIL-seq data sets (18 in total).

As suggested, we now provide more details about the different RIL-seq datasets we used in the analysis (page 10, lines 202-205).

It would be helpful to specify the criteria for "top" interactions in targets retrieved from RIL-seq data (Table S1 and text, e.g., line 273): e.g. number of conditions, number of chimeras, etc.

As suggested, we now more explicitly specify the criteria for selecting targets to characterize (page 10, lines 205-206).

Fig. 4B/ S6 and line 242: The flagellin amount in the empty vector control (pZE) looks higher than in WT, and the stated effect of MotR/MotR* OE on flagellin is not very clear from the blot. The "cross-reacting band" above flagellin also seems to vary among strains. Could the authors include a quantification of flagellin protein amount and normalize relative to a housekeeping protein (e.g., GroEL), instead of Ponceau S as loading control?

We agree that there is some variation between the levels of flagellin in the WT and pZE control sample, possibly due to the addition of antibiotic to the pZE culture. We added quantification of the bands in Figure 4—figure supplement 1 to better document the changes in flagellin levels.

Figure legends: It would be helpful to have a bit more information about the method used/displayed image rather than stating results in the legends.

As suggested, we now provide a bit more information about the methods used/displayed image in the figure legends to allow for easier comprehension of the data presented in the figures (while trying to balance this with the length of the legends).

Fig. 2: Please include a scale for all electron microscopy images or, if it is the same for all panels, state it in the figure legend. Moreover, the same image is used for the pZE control in panel C, E and Figure S4A/C. It would be better to show different fields of bacteria for the pZE sample.

As is now mentioned in the legends to Figure 2, Figure 2—figure supplement 2, and Figure 8, the same scale was used for all panels. We thought it was better to show the same image for the pZE control in the different panels to emphasize that these samples were all analyzed on the same day.

Fig. 2: The sRNA OE strains seem to show some heterogeneity in cell length (pZE-MotR) or width (pZE-FliX). The authors could, e.g., check whether this is a phenotype correlated to sRNA OE by quantifying these parameters for different fields and comparing to WT or comment on this in the text if this is not consistently seen.

We also were intrigued by the slightly different sizes and widths of cells in the EM images. However, our statistical analysis did not reveal significant differences between the different samples. We now comment on this (page 53, lines 1178-1179).

As a follow-up to this study, it would be interesting to assess the impact of MotR and FliX regulation of ribosomal protein synthesis on overall ribosome activity (e.g., via Ribo-seq), also considering that antitermination regulates rRNA transcription. In the case of MotR, the authors suggest that MotR upregulation of S10 protein might not only impact antitermination, but also lead to the formation of more active ribosomes that would increase flagellar protein synthesis (lines 359-362). However, in the RNA-seq performed in OE MotR* several transcripts encoding rRNA and ribosomal proteins are significantly downregulated compared to EVC (Supplementary Table S2). Could the authors comment on this?

We share the reviewer’s enthusiasm for follow-up work and thank for the suggested experiments. We hope we will be able to decipher the full mechanism of MotR and FliX action on ribosomal protein synthesis in future experiments. The observation that some ribosomal protein-coding gene levels are reduced in the RNA-seq experiment with overexpression of MotR* is interesting but we do not have an explanation other than the fact that the samples were collected early in exponential growth. We now mention the observation in the text (page 19, lines 404-407).

Considering that OE of the WT MotR appears to increase fliC mRNA abundance but has no strong impact on flagellin protein levels, can the authors speculate what is the physiological relevance of MotR* for flagellin production?

We agree that while we do see significant increases in the flagella number and fliC mRNA abundance with MotR and MotR* overexpression, the western analysis did not reveal a striking increase in flagellin levels and also wonder how MotR strongly increases the flagella number, which requires flagellin subunits, but only has a weak effect on the intercellular levels of flagellin. One possibility explanation is that it is more difficult to see significant increases for a protein whose levels are high to begin with. These points are now discussed (page 13, lines 264-269).

Fig. 4C: The pZE samples seem to show variable expression of fliC mRNA although the samples are collected at the same timepoints. Try to clarify in the text.

The northern membrane on the bottom was exposed for a longer time due to the lower fliC mRNA levels in the samples with FliX overexpression. We now note these differences in the legends to Figure 4 and Figure 4—figure supplement 1.

Fig. 7/S13: While a volcano plot for MotR* is shown in Fig. 7A, quantification of GFP reporter fusion regulation is shown for MotR. Quantifications of MotR* are shown in Fig. S13. Maybe swap the figures.

Given that the data for MotR* are in the supplement figures for all other figures we would also like to retain this distribution for Figure 7 (aside from the volcano plot since this experiment was only carried out for MotR*).

Lines 135-136 (Fig. S1B): on the northern blots, only sRNA levels of MotR are comparable between rich and minimal media (excluding M63 G6P and M63 gal). Most other sRNA seem to be more abundantly expressed in minimal media conditions compared to LB. Maybe rephrase.

As suggested, the text was revised to point out the differences in the sRNA levels for cells grown in different growth media (page 7, lines 140-144).

Lines 229-234: this paragraph seems not directly connected to the aims of the study (i.e., no effect on motility tested of these other sRNAs) and could be removed (or moved to discussion).

We appreciate the reviewer’s suggestion but, considering Reviewer 1’s comments, think that showing the regulation of lrhA by other sRNAs has value in highlighting the complexity of the regulatory circuit. We have revised the text to incorporate Reviewer 1’s suggestions and better explain why these results are intriguing (page 12, lines 247-250).

Line 200 and Fig. S5: For FlgO sRNA only one target was identified in RIL-seq. This gene could be specified and labeled in Fig. S5 and the text. Does FlgO also bind ProQ?

We now mention the single FlgO target (gatC) detected in four datasets (page 10, lines 213215). In Figure 3—figure supplement 1, we labeled only targets that we followed up with in the current study. Therefore, to be consistent, we prefer not to label gatC in the FlgO plot. FlgO was found to co-immunoprecipitate with ProQ but at much lower levels than with Hfq, and to have very few RNA partners (Melamed et al., 2020).

Lines 493-498: It is mentioned that the four sRNAs were also detected in recent RIL-seq experiments of *Salmonella* and EPEC. Are any of the here identified targets also found in other species or was none detected as analyses were carried out under conditions that do not favor flagella expression?

The targets identified in this study were not detected in the *Salmonella* and EPEC RIL-seq datasets. However, the *Salmonella* and EPEC experiments were carried out under different growth conditions. Based on the sequence conservation of the Sigma 28-dependent sRNAs across several bacterial species (Figure 8—figure supplement 2), we do think overlapping targets will be found in other bacterial species under the appropriate growth conditions.

The strongest evidence of MotR dependent target regulation is the one on rpsJ, which does not necessarily require the additional experiments with MotR*. Since the authors were able to show upregulation of the rpsJ-gfp reporter upon OE of MotR WT, it would have strengthened the results if they performed the experiments in Fig. S8C with MotR WT. Similary as an increase of flagella number was seen with OE of MotR WT in Fig. 2A, the effect of the OE S10∆loop could be compared to OE MotR instead of OE MotR* (Fig. 6A). At least if would be helpful, to briefly comment on why MotR* was used instead of MotR WT for these experiments.

As suggested, we state MotR* was used in some assays given the stronger effects for some phenotypes (page 10, lines 196-197). We think, given that we established MotR and MotR* cause the same effects, with increased intensity for the latter, it is reasonable to use MotR* in some of the experiments.

p. lines 482-491 and 508-511: The authors discuss that both UhpU sRNAs and RsaG sRNA from *S. aureus* are derived from the 3'UTR of uhpT, but conclude there is no overlap regarding flagella regulation, suggesting independent evolution of these sRNAs. However, the authors also mention that UhpU sRNA has many additional targets beyond LhrA involved in carbon and nutrient metabolism. Thus, maybe regulation of metabolic traits could be a conserved theme and function for UhpU and RsaG? Maybe try to comment on or better connect these two parts in the discussion.

As suggested, we now comment on the possibility of the regulation of metabolic traits being a conserved theme and function for UhpU and RsaG (page 24, lines 520-527).

Check the text for consistency regarding the use of italics for gene names (e.g., legend of Figs. 7 and 8)

The text was corrected.

Please introduce abbreviations, e.g., G6P (line 139), REP (line 150), ARN (line 258), NOR/U (Table S1 legend)

As suggested, we now introduce the abbreviations for G6P (page 7, line 142), REP (page 8, lines 155-156), and NOR (Supplementary file 1 legend). Regarding ARN, these sequences are already written in parentheses in the same sentence. However, we revised this to “ARN motif sequences” (page 13, line 278).

Fig. S1A: Highlight REP sequence mentioned in text (line 150).

REP sequences are now highlighted in gray in Figure 1—figure supplement 1A.

Fig. S1C: It would be helpful to list number nt positions on the sRNAs based on full-length transcripts.

The corresponding positions based on the full-length transcripts have also been added to this figure.

Fig. S2: Adjust the position of UhpU-S label.

UhpU-S label position was adjusted.

Fig. S6: Include UhpU in the figure title.

UhpU was added to the title.

Fig. S10: It would be helpful to indicate on the figure (or state more clearly in the legend) which RNA was extracted from WT or ΔfliCX background.

The samples shown in the Figure are all in a WT strain. We corrected the figure legend accordingly.

Line 290: the effect is on flagella number, not motility.

This typo is now corrected (page 15, line 312).

Fig. S8: One-way ANOVA (panel A legend)

This typo is now corrected (page 64, line 1433).

Line 320: Fig. S9C instead of 9C

We thank the reviewer for noticing the typo. The numbering of the supplementary figures has now been changed to the eLife format.

It would be helpful to add reference for statement in line 57.

A reference to (Fitzgerald et al., 2014) was added as suggested.

Add PMID:32133913 as reference for post-transcriptional regulation of the flagellar regulon in the introduction (lines 87-91)

The indicated reference was added as suggested (page 5, lines 87-91).

Legend Fig. S6: expand view -> expanded view

This typo is now corrected (page 63, line 1406).

line 513: sRNA -> sRNAs

This typo is now corrected (page 25, line 549).

Fig. 8G: Maybe include lrhA as target of UhpU sRNA at top of the cascade.

As suggested lrhA has been added as a target of UhpU at the top of the cascade.